# Exploring Evolutionary Pathways and Abiotic Stress Responses through Genome-Wide Identification and Analysis of the Alternative Oxidase (AOX) Gene Family in Common Oat (*Avena sativa*)

**DOI:** 10.3390/ijms25179383

**Published:** 2024-08-29

**Authors:** Boyang Liu, Zecheng Zhang, Jinghan Peng, Haipeng Mou, Zhaoting Wang, Yixin Dao, Tianqi Liu, Dandan Kong, Siyu Liu, Yanli Xiong, Yi Xiong, Junming Zhao, Zhixiao Dong, Youjun Chen, Xiao Ma

**Affiliations:** 1College of Grassland Science and Technology, Sichuan Agricultural University, Chengdu 611130, China13008108360@163.com (H.M.);; 2College of Grassland Resources, Southwest Minzu University, Chengdu 610041, China

**Keywords:** AOX gene family, *Avena sativa*, evolutionary pathways, abiotic stresses

## Abstract

The alternative oxidase (AOX), a common terminal oxidase in the electron transfer chain (ETC) of plants, plays a crucial role in stress resilience and plant growth and development. Oat (*Avena sativa*), an important crop with high nutritional value, has not been comprehensively studied regarding the AsAOX gene family. Therefore, this study explored the responses and potential functions of the AsAOX gene family to various abiotic stresses and their potential evolutionary pathways. Additionally, we conducted a genome-wide analysis to explore the evolutionary conservation and divergence of AOX gene families among three *Avena* species (*Avena sativa*, *Avena insularis*, *Avena longiglumis*) and four *Poaceae* species (*Avena sativa*, *Oryza sativa*, *Triticum aestivum*, and *Brachypodium distachyon*). We identified 12 AsAOX, 9 AiAOX, and 4 AlAOX gene family members. Phylogenetic, motif, domain, gene structure, and selective pressure analyses revealed that most AsAOXs, AiAOXs, and AlAOXs are evolutionarily conserved. We also identified 16 AsAOX segmental duplication pairs, suggesting that segmental duplication may have contributed to the expansion of the AsAOX gene family, potentially preserving these genes through subfunctionalization. Chromosome polyploidization, gene structural variations, and gene fragment recombination likely contributed to the evolution and expansion of the AsAOX gene family as well. Additionally, we hypothesize that *AsAOX2* may have potential function in resisting wounding and heat stresses, while *AsAOX4* could be specifically involved in mitigating wounding stress. *AsAOX11* might contribute to resistance against chromium and waterlogging stresses. *AsAOX8* may have potential fuction in mitigating ABA-mediated stress. *AsAOX12* and *AsAOX5* are most likely to have potential function in mitigating salt and drought stresses, respectively. This study elucidates the potential evolutionary pathways of the AsAOXs gene family, explores their responses and potential functions to various abiotic stresses, identifies potential candidate genes for future functional studies, and facilitates molecular breeding applications in *A. sativa*.

## 1. Introduction

Throughout their lifecycle, higher plants face a variety of abiotic stresses, such as chromium contamination, extreme temperatures, waterlogging, and pest damage, which can lead to irreversible cellular damage and significant yield losses [1,2,3,4,5,6,7]. Abscisic acid (ABA), a key plant hormone, plays a crucial role in regulating many physiological responses to these stresses. To mitigate these challenges, plants have evolved complex mechanisms involving gene expression and regulatory networks [3,8]. Key gene families, including trihelix, HSP, glyoxalase, and AOX, are essential for stress resistance [4,9,10,11]. Additionally, plant respiration is tightly controlled by metabolic networks that allow plants to adjust their metabolism in response to internal cellular demands and external environmental stresses. Notably, higher plants, algae, and most fungi possess an alternative respiratory pathway in their mitochondria, which is resistant to cyanide and bypasses the cytochrome pathway, ultimately reducing oxygen to water [12]. The alternative oxidase (AOX) is a key component of this pathway and is critical for its proper function under various stress conditions.

AOX is a redox enzyme located at the end of the Electron Transfer Chain (ETC) (Figure 1), identified in all surveyed plants [13]. AOX is an integral membrane protein located in the mitochondrial inner membrane. It can regulate the levels of potential mitochondrial signaling molecules, such as superoxide, nitric oxide, and key redox pairs [14]. In higher plants, AOX is encoded by a small family of nuclear genes. AOX notably bypasses complexes III and IV, consuming excess electrons and reducing the production of reactive oxygen species (ROS), thereby mitigating oxidative stress in plant cells and contributing to resistance against various environmental stresses [5,15,16,17]. Therefore, AOX genes respond to various stress conditions, particularly those that affect ROS levels, such as changes in temperature and light intensity, as well as salt, drought, and osmotic stress [18,19,20,21]. The expression of AOX genes is associated with both plant growth and development (such as early cell reprogramming and leaf growth) and stress induction [18,19,22]. Additionally, some transcription factors (e.g., WRKY, NAC, bZIP, and MYB) may regulate AOX gene responses to various environmental stresses [18,23]. Due to its involvement in nearly all types of stress responses, the AOX gene is considered a valuable marker for stress tolerance in crop breeding, aiming to produce plants with high resilience to various environmental conditions [24]. The AOX gene family comprises two subfamilies: AOX1 and AOX2 [11,18,25]. AOX1 is found in multiple dicotyledonous and monocotyledonous plants, whereas AOX2 appears to have been lost during the evolution of monocotyledon species [26]. The AOX1 subfamily is represented by multiple gene copies in many plants, while the AOX2 subfamily shows relatively fewer expansions [27]. The generation and diversification of AOX1 and AOX2 subfamilies likely occurred during the early evolution of angiosperms [28].

Gene duplication creates new genes, providing new functions and promoting the evolution and expansion of gene families. The evolution of gene families is fundamental to species evolution and is likely closely related to mechanisms of environmental adaptation in different species [29,30]. Oat (*A. sativa*) is an important crop in the *Poaceae* family, recognized as a low-carbon crop with high nutritional value [31]. *Avena* exists in nature as diploid, tetraploid, and hexaploid. Studies have shown that hexaploid *A*. *sativa* originated from the hybridization between tetraploid *A. insularis* and diploid *A*. *longiglumis*, which are potential ancestral species sharing many homologous genes with *A. sativa*. [31]. AOX genes have been extensively studied in animals [32], fungi [33], and higher plants including *Oryza sativa* [34], *Glycine max* [35], *Zea mays* [36], *Nicotiana tabacum* [37], *Triticum aestivum* [38,39], *Arabidopsis thaliana* [40], *Hypericum perforatum* [41], *Vitis vinifera* [27], *Citrus clementina*, *Citrus sinensis* [42], and *Phyllostachys edulis* [11]. However, comprehensive studies on the AOX gene family in oats remain relatively scarce. High-quality whole genome sequencing and assembly of *A. sativa*, *A. insularis*, and *A. longiglumis* facilitate comprehensive studies of the AsAOX, AiAOX, and AlAOX gene family at the genomic level [31,43]. Previous studies have extensively explored the responses and potential functions of the AOX gene family to salt and drought stresses [11,18,19,21,38]. However, little is known about their responses and potential functions to novel abiotic stresses, such as Cr stress, wounding stress, and waterlogging stress. Therefore, this study aims to explore the responses and potential functions of the AsAOX gene family under various abiotic stresses (heat stress, wounding stress, waterlogging stress, ABA-mediated stress, Cr stress, salt stress, and drought stress) using RT-qPCR and RNA-Seq. Moreover, we employed multiple methods to comprehensively identify and analyze the AOX gene family at the genome-wide level in *A. sativa*, *A. longiglumis*, and *A. insularis*, ultimately describing the potential evolutionary pathways of the AsAOX gene family. It also serves as a scientific reference for subsequent studies on gene function verification and the elucidation of regulatory mechanisms.

## 2. Results

### 2.1. Identification and Chromosomal Location of AsAOXs, AlAOXs and AiAOXs

We identified 12 AsAOX members in *A. sativa* (AACCDD), 4 AlAOX members in *A. longiglumis* (AA), and 9 AiAOX members in *A. insularis* (CCDD) (Figure 2). In *A. sativa*, the 12 AsAOX genes are distributed across Chr2A, Chr2C, Chr2D, Chr5D, Chr6A, Chr6C, and Chr6D. In *A. insularis*, the 9 AiAOX genes are located on Chr2C, Chr2D, Chr4C, Chr5D, Chr6C, and Chr6D. The 4 AlAOX genes in *A. longiglumis* are found on Chr2A and Chr6A. We artificially renamed members of the AsAOX, AlAOX, and AiAOX gene members as *AsAOX1*~*AsAOX12*, *AlAOX1*~*AlAOX4*, and *AiAOX1*~*AiAOX9* (Appendix A).

Based on chromosome localization, we observed that certain genes in *A. sativa* correspond to genes in their ancestral species. For instance, *AsAOX7* and *AiAOX8* are located on Chr5D, *AsAOX11* and *AiAOX4* on Chr6C, and *AsAOX12* and *AiAOX3* on Chr6D. Gene clusters are present on multiple chromosomes, with some clusters being relatively conserved across the three species. Additionally, in *A. insularis*, an AiAOX gene is located on Chr4C, whereas no corresponding AsAOX gene is found on Chr4C of *A. sativa*. Frequent translocations and inversions during *A. sativa*’s polyploidization events may be associated with this phenomenon [43,44].

### 2.2. Physicochemical Properties and Subcellular Localization Prediction of AsAOXs, AlAOXs, and AiAOXs

We used TBtools-II v2.119 and WoLF PSORT (https://wolfpsort.hgc.jp/, accessed on 5 February 2024)to analyze amino acid counts (AA), molecular weight (MW), isoelectric point (PI), instability index (II), aliphatic index (AI), and Grand average of hydropathicity (GRAVY) for 25 AOX protein sequences from *A. sativa*, *A. longiglumis*, and *A. insularis*, as well as to predict their subcellular localization (SL). The physicochemical analysis revealed that AsAOXs have AA counts ranging from 319 to 416, MW from 36,735.62 to 46,772.25, PI from 5.01 to 9.3, II from 32.25 to 68.02, AI from 71.31 to 88.33, and GRAVY from −0.344 to −0.115. AiAOXs have AA counts ranging from 312 to 385, MW from 35,018.92 to 42,550.74, PI from 5.24 to 9.39, II from 32.5 to 65.94, AI from 70.57 to 88.63, and GRAVY from −0.399 to −0.346. AlAOXs have AA counts ranging from 320 to 353, MW from 36,567.26 to 40,197.49, PI from 4.96 to 8.84, II from 37.72 to 62.99, AI from 73.03 to 88.78, and GRAVY from −0.295 to −0.102 (Table 1).

Additionally, AsAOXs display broader ranges in AA and MW compared to AiAOXs and AlAOXs. The upper limit of PI values for AlAOXs is notably lower than for the other two, likely due to *A. longiglumis* being diploid. All three gene families exhibit AI values above 70, indicating a preference for hydrophobic environments. Subcellular localization prediction for AsAOXs, AiAOXs, and AlAOXs suggest that, except for *AsAOX2*, which may localize in the nucleus or mitochondria, the remaining 24 members are predominantly found in mitochondria or chloroplasts (Table 1).

### 2.3. Phylogenetic Analysis of AOX Genes in Oat and 10 Plants

To understand the evolutionary relationships of AOX gene family members in *A. sativa* and several monocot and dicot plants, we constructed a Maximum Likelihood (ML) phylogenetic tree using 81 renamed AOX protein sequences in MEGA v7.0, divided into five clusters (Figure 3 and Appendix A). Cluster 1 contains the highest number of AOX genes (31), including all HvAOX (4/4) and BdAOX (6/6), most AsAOX (9/12), ZmAOX (5/6), OsAOX (4/5), and a few SbAOX (3/8). This suggests that Cluster 1 likely underwent multiple gene duplication events, resulting in a large number of duplicated genes [45,46]. Cluster 5 also contains a substantial number of AOX members (26), including all TaAOX (18/18), ScAOX (3/3), and most SbAOX (5/8). Cluster 2 includes all AtAOX (5/5), most MtAOX (4/5), and a few GmAOX (4/9). Cluster 3 contains fewer MtAOX (1/5), OsAOX (1/5), ZmAOX (1/6), BdAOX (1/6), and AsAOX (3/12). Additionally, Cluster 4 exclusively contains GmAOX. In the phylogenetic tree, AOX genes from the same species often cluster together (e.g., HvAOXs, BdAOXs, TaAOXs), indicating strong evolutionary conservation in these species. Notably, except for a few AOX genes, most monocot and dicot AOX genes cluster together in the same clusters, broadly aligning with the classification of monocots and dicots across different species. Interestingly, we observed that, in the ML phylogenetic tree, AsAOX clusters not only with HvAOX but also shows high homology with BdAOX. This may result from similar evolutionary pressures among these three species. Overall, most AOX genes across the 11 species exhibit a certain degree of evolutionary conservation. However, some AOX genes may have evolved different functions adapted to environmental conditions, exhibiting distinct clustering from most family members and indicating potential roles in species-specific adaptation.

### 2.4. Evolutionary Conservation Analysis of AsAOXs, AiAOXs, and AlAOXs

To further understand the evolutionary conservation of AOX genes in *A. sativa*, *A. longiglumis*, and *A. insularis*, we conducted a comprehensive analysis of 25 AOX protein sequences, including phylogenetic analysis, conserved motif analysis, conserved domain analysis, and gene structure analysis (Figure 4). The results classified these sequences into three clusters represented by blue, green, and pink backgrounds. Importantly, the distribution of AOX genes was uniform in different clusters. Phylogenetic analysis revealed high homology between several AiAOX and AlAOX genes with AsAOX genes (Figure 4A), consistent with the hybridization of *A. longiglumis* and *A. insularis* resulting in *A. sativa*.

Furthermore, based on conserved motif analysis, all genes in each cluster contained motif2~motif5, with *AiAOX9* in Cluster 3 having the fewest motifs (4–6), while members in Clusters 1 and 2 possessed the most motifs (8) (Figure 4B). Conserved domain analysis identified PLN02478 and AOX domains in AOX proteins across all three species, with high similarity in domain types and positions among members within the same cluster (Figure 4C). Different domains likely correspond to distinct biological functions, suggesting variability in functional roles between members of Clusters 1, 2, and 3. Notably, genes identified as AOX family members based on BLASTP and HMM Search in *A. sativa*, *A. insularis*, and *A. longiglumis* typically contained the ‘PLN02478’ domain, implying that this domain may represent a branch of the AOX gene family or share a common ancestor during evolution. For a deeper understanding of the conservation of AOX gene structures, we performed AOX gene structure analysis across the three species using Tbtools-II v2.119 (Figure 4D). Introns can influence gene recombination and alternative splicing of ncRNAs, thereby impacting gene evolution [47]. Notably, *AsAOX8* in Cluster 1 exhibited the longest UTR, while *AsAOX4* in Cluster 2 had the longest intron. These structural differences from other AOX genes within the same cluster suggest potential new gene structure variations arising during AsAOX gene family formation and evolution. The five members in Cluster 1 displayed relatively consistent gene structure distribution, indicating potential functional conservation among these genes. Moreover, Clusters 1 and 3 exhibited relatively longer or more numerous introns, suggesting possible events of intron insertion or expansion during evolution that could affect gene expression [48]. Additionally, some genes showed UTR deletions (e.g., in Cluster 3), potentially indicating adaptive evolutionary responses to environmental factors.

Overall, the comprehensive analysis revealed substantial similarities in motif recognition, conserved domain positions and types, and gene structure within the same cluster of AOXs in three *Avena* species, suggesting high functional similarity and evolutionary conservation among these proteins. Conversely, differences in motif count, domain types, and gene structure, particularly among Cluster 3 AOX members, hint at potential evolutionary divergence and distinct biological functions for these genes compared to other clusters.

### 2.5. Synteny and Orthologous Genes Analysis of AOXs Genes

Synteny analysis reflects the conservation of gene order, sequence, and position across different species. If genes in different species exhibit synteny relationships, they are likely to perform similar biological functions [49]. To assess the evolutionary relationships of AsAOX with nine AOX gene families in other species, we conducted synteny analysis across these 10 plant species (Figure 5). However, we found no synteny gene pairs between AsAOX and AtAOX (*Arabidopsis thaliana)*, suggesting AOX genes may be involved in the differentiation of dicotyledonous plants. Therefore, the synteny maps of *A. sativa* with the remaining eight species were used for subsequent analysis (Figure 5 and Appendix A). Synteny analysis revealed that six AsAOX genes (*AsAOX1*, *AsAOX3*, *AsAOX6*, *AsAOX8*, *AsAOX11*, *AsAOX12*) in Cluster 1 have syntenic relationships with AOX genes from other species. other species. Furthermore, to explore the homology of AsAOX with AiAOX and AlAOX genes, we used OrthoVenn3 to identify orthologous genes of the 12 AsAOX genes in the full-length protein sequences of *A. insularis* and other species (Appendix A).

### 2.6. Analysis of Gene Duplication and Selection Pressure of AsAOX, AiAOX, and AlAOX

To understand the expansion/contraction of the AsAOX, AiAOX, and AlAOX gene family during evolution, we identified gene duplication events using Tbtools-II v2.119 (Figure 6). According to the gene duplication analysis, a total of 9 out of 12 AsAOX genes were involved in gene duplication events, resulting in 16 paralogous gene pairs (Table 2). Additionally, we identified seven AiAOX and one AlAOX paralogous gene pairs. Furthermore, some AsAOX genes underwent multiple duplication events and showed paralogous relationships with genes from the same or different subgenomes (e.g., *AsAOX1*), suggesting these genes played a crucial role in the expansion of the AsAOX gene family.

Additionally, to gain deeper insights into the evolutionary process of AsAOX, AiAOX, and AlAOX paralogous gene pairs, we computed Ka, Ks, and ka/ks values using Tbtools-II v2.119 (Table 2). Ka/Ks ratios are used to estimate evolutionary selection pressure, where values greater than 1, less than 1, and equal to 1 indicate positive selection, purifying selection, and neutral selection, respectively. Evolutionary selection analysis revealed that all paralogous gene pairs had Ka/Ks ratios less than 1, indicating that the AsAOX, AiAOX, and AlAOX gene family primarily underwent purifying selection during evolution.

### 2.7. Analysis of Cis-Regulatory Elements of AsAOXs

To understand the gene expression regulation mechanisms of AsAOXs, we predicted cis-regulatory elements (CREs) in the upstream 1500 bp sequences of 12 AsAOX genes (Figure 7B and Appendix A). Additionally, to explore the phylogenetic relationships among these 12 AsAOX genes, we constructed a maximum likelihood (ML) evolutionary tree using MEGA v7.0 based on 1000 bootstrap values (Figure 7A). CREs analysis revealed that eukaryotic promoter elements such as CAAT-box and TATA-box were widely distributed in the promoter regions of AsAOXs, along with numerous CREs related to abiotic and biotic stress responses, phytohormone responsiveness, and plant growth and development [50]. *AsAOX8* exhibited the highest number of CREs (44), encompassing diverse types of CREs, suggesting its pivotal role in stress response, hormone signaling, and developmental regulation. Notably, *AsAOX11* possessed the highest number of stress-responsive CREs (21), indicating its potential role in stress responses. Moreover, within Cluster1, *AsAOX3* had the highest number of hormone-responsive CREs (21), highlighting its potential role in hormone signaling. Furthermore, stress-responsive CREs (136, 35.5%) and hormone-responsive CREs (147, 38.4%) were both more abundant than growth and development-related CREs (100, 26.1%) among the 12 AsAOX genes, suggesting a potential emphasis of AsAOX genes in stress and hormone responses.

To further understand the differences in the quantity and types of CREs in the promoter regions of AsAOXs, AiAOXs, and AlAOXs, and their relationship to evolution, we predicted CREs in the promoter regions of AiAOX and AlAOX (Figure 7C and Appendix A). New CREs are sources of new traits and form the basis of evolution. We found that, compared to AiAOX and AlAOX, AsAOX exhibited new CREs related to hormone responses. The types of stress-related CREs in AsAOX were generally similar to AiAOX and AlAOX (10), while the number of stress-related CREs in AsAOX (136) was approximately equal to the combined total of AiAOX (106) and AlAOX (42), suggesting higher conservation of stress-responsive CREs during the evolutionary process of the AOX gene family in *A. insularis*, *A. longiglumis*, and *A. sativa*.

### 2.8. Analysis of 3D Structure of AsAOXs Proteins

AsAOX proteins play crucial roles in *A. sativa*, and protein function is closely associated with protein 3D structure. Homology modeling operates on the premise that, if two proteins share high sequence similarity, their 3D structures are likely similar, facilitating prediction of unknown protein structures based on known ones. In this study, homology modeling using AlphaFold v2 predicted the three-dimensional structures of 12 AsAOX proteins (Appendix A). Different colors represent distinct internal protein structures: purple, blue, cyan, white, and transparent shells correspond to α helices, β turns, β sheets, irregular coils, and protein shells, respectively. Models were evaluated using SWISS-MODEL and Ramachandran Plot assessments (Appendix A), showing sequence identities ranging from 73.16% to 89.5%, GMQE scores between 0.7 and 0.80 (except for AsAOX2, 0.61), indicating generally satisfactory model quality for most AsAOX proteins. Additionally, Ramachandran Plot assessments indicate that the 3D structural quality of 12 AsAOX proteins is satisfactory. Notably, all AsAOX proteins exhibited one or more prominent alpha helices, underscoring the significant role of this secondary structure in AsAOX protein functionality.

### 2.9. Analysis of Gene Ontology (GO) Enrichment and Protein–Protein Interaction Network

The enrichment analysis of AsAOX genes reveals that all AsAOX genes are enriched in five molecular functions (MFs), two cellular components (CCs), and three biological processes (BPs). These results suggest that AsAOX gene members likely play roles across three functional categories, with a predominant involvement in molecular functions. All AsAOX genes are enriched in GO:0009916 (alternative oxidase activity) (Appendix A), indicating that they possess alternative oxidase activity. However, Cluster3 (AsAOX7, AsAOX10, AsAOX9) did not enrich in one cellular component (GO:0005739) and five biological processes (GO:0009060, GO:0015980, GO:0010230, GO:0045333, GO:0006091), indicating functional differences between Cluster 3 and other clusters; the result is consistent with the domain analysis result (Figure 4B and Figure 8).

To further investigate the interaction between AsAOX proteins and other proteins, we conducted PPI predictions (Appendix A). Interactions between different proteins suggest that they might work together to perform specific functions. A total of five proteins have strong interactions with AsAOX proteins: A0A3B5ZNL9, A0A3B6H2Q9, A0A3B6EPQ1, A0A3B6FXG7, and A0A341NWK8. These proteins all belong to the Casparian strip membrane proteins (CASPs) family. The strong interaction between CASP and AOX proteins in the PPI network suggests that they may have synergistic or complementary functions in root metabolism, structural regulation, environmental stress response, and signal transduction.

### 2.10. Expression Analysis and Tissue Specificity of Leaf and Root by RT-qPCR

The AOX gene family plays a crucial role in plant stress resistance. To explore whether the identified 12 AsAOX gene members could respond to various abiotic stresses, we conducted RT-qPCR analyses on root and leaf tissues under waterlogging, chromium (Cr), wounding, ABA, and heat stress conditions (Figure 9 and Appendix A). In oat, the majority of AsAOX genes in both roots and leaves responded to these five abiotic stresses, with the response to heat stress being the most pronounced. In leaves, the highest gene expression levels reached approximately 210-fold of the control (CK) for *AsAOX2* and *AsAOX11*, while, in roots, the expression peaked at around 30-fold of the CK for *AsAOX10*.

In leaves, ABA treatment notably induced the expression of *AsAOX2*, *AsAOX4*, and *AsAOX6*, reaching maximum transcription levels at 24 h and 48 h. Under heat stress, the expression of *AsAOX2* and *AsAOX11* was particularly dramatic, peaking at 24 h and 9 h, respectively. For wounding stress, *AsAOX2* and *AsAOX4* showed notable expression levels, peaking at 3 h and 48 h. Under waterlogging stress, *AsAOX2* and *AsAOX4* also exhibited significant expression levels, peaking at 6 h and 3 h. Under Cr stress, only *AsAOX2* and *AsAOX12* showed significant gene expression levels, peaking at 24 h and 3 h, respectively. In oat leaf tissues, most AsAOX genes could respond to these five abiotic stresses, with *AsAOX2* showing a distinct response to all stresses.

In root tissues, *AsAOX8* and *AsAOX12* notably responded to ABA-mediated stress, reaching peak expression levels at 3 h and 6 h, respectively. Additionally, *AsAOX2*, *AsAOX5*, *AsAOX8*, *AsAOX10*, and *AsAOX12* showed significant responses to heat stress, with peaks at 24 h, 24 h, 48 h, 48 h, and 24 h, respectively. *AsAOX2*, *AsAOX4*, and *AsAOX12* significantly responded to wounding stress, peaking at 3 h, 3 h, and 6 h. For waterlogging stress, *AsAOX9* and *AsAOX12* displayed notable gene expression levels, peaking at 48 h and 24 h. Under Chromium stress, *AsAOX2*, *AsAOX8*, and *AsAOX12* had significant gene expression levels, peaking at 24 h, 6 h, and 24 h. Notably, *AsAOX12* showed notable responses to all five abiotic stresses in both root and leaf tissues. Moreover, *AsAOX11* exhibited very-low gene expression levels in roots under all five stress conditions.

### 2.11. Oat Expression Analysis of Leaf by RNA-Seq

We utilized publicly available oat transcriptome data to further explore the expression patterns of 12 AsAOX genes under salt and drought stress (Figure 10). *AsAOX7* and *AsAOX12* showed significant responses to salt stress, whereas *AsAOX2* and *AsAOX6* exhibited lower responses to salt stress. Under drought stress, *AsAOX5* and *AsAOX7* were notably downregulated, while the other genes showed upregulation.

## 3. Discussion

### 3.1. Phylogenetic Analysis

The alternative oxidase (AOX) protein, a type of terminal oxidoreductase located at the end of the electron transport chain, has been identified across various plant species and responds to multiple abiotic stresses [15,16]. AOX is crucial for the normal function of the alternative respiration pathway under environmental stress conditions. This pathway helps maintain cellular redox balance, reduces the generation of reactive oxygen species (ROS), maximizes photosynthetic efficiency, and ensures the continuous operation of the electron transport system (ETS) in the presence of inhibitors and environmental stresses [15,17]. Given the evolutionary principles, we utilized Maximum Likelihood (ML) analysis of AOX protein ML trees from eight monocot and three dicot plant species to explore the evolutionary relationships of the AOX gene family [51,52] (Figure 3). We observed significant differences in the number of AOX gene members across various clusters, indicating a high degree of variability in the number and family expansion of AOX genes (ranging from 4 to 17) among these 11 species [41]. The AOX gene family has the fewest members in *Secale cereale* and *A. longiglumis*, but the most in *Triticum aestivum* and *A. sativa*. This suggests a strong correlation between the number of AOX gene family members and the polyploidy and size of their genomes [10] (Figure 3). Based on their evolutionary branches, these were divided into five clusters. Cluster 1, containing nine AsAOX genes, had the highest number of AOX genes, with several exhibiting multiple segmental duplications (Figure 6). This suggests that multiple gene duplication events may have occurred in Cluster 1 [45,46]. Additionally, AsAOX genes not only clustered on the same evolutionary branch as HvAOX but also showed high homology with BdAOX, possibly due to similar selective pressures faced by these AOX gene families during evolution. Furthermore, most AOX genes from the 11 species showed evolutionary conservation and clustering consistent with the classification into monocots and dicots. However, a few genes, such as *AsAOX7*, *AsAOX9*, and *AsAOX10*, did not cluster with the majority of AOX genes within the same family, likely due to differences in their protein domains and distinct evolutionary pathways (Figure 3 and Figure 4).

### 3.2. Exploration of Evolutionary Pathways

Gene family evolution often accompanies gene formation and expansion, with tandem and segmental duplications widely recognized as promoters of gene family expansion [5,9,18]. In this study, 9 out of the 12 AsAOX genes participated in segmental duplication, forming 16 segmental duplication gene pairs, with no tandem duplication identified (Figure 5 and Table 2). These segmental duplication events contributed to the formation and expansion of the AsAOX gene family [53], aligning with previous research [11,23]. Additionally, we found that the paralogous gene pairs of the AsAOX, AiAOX, and AlAOX gene families have Ka/Ks values all less than 1, while the paralogous gene pairs of the PeAOX gene family also exhibit Ka/Ks values less than 1 [11] (Table 2). Therefore, the AOX gene families are likely under purifying selection across multiple species during evolution, indicating strong conservation [49,52]. Interestingly, no syntenic relationships were found between AsAOX and AtAOX genes, suggesting the potential involvement of the AOX gene family in the differentiation of dicots. Additionally, *AsAOX1*, *AsAOX3*, *AsAOX6*, *AsAOX8*, *AsAOX11*, and *AsAOX12* (Cluster 1) exhibited synteny relationship with most plant genomes used in this study, highlighting its possible critical role in the evolution of the AOX gene family [49] (Figure 5 and Appendix A). The “A” subgenome and “CD” subgenome in hexaploid *A. sativa* (AACCDD) are derived from its potential ancestral species *A. longiglumis* (AA) and *A. insularis* (CCDD) [43](Figure 2 and Appendix A). The combined AOX gene count in *A. longiglumis* and *A. insularis* approximates the AsAOX gene count in *A. sativa*, providing evidence that *A. longiglumis* and *A. insularis* are the potential ancestral species of *A. sativa* [24,37]. This finding aligns with previous research on AsHSP90 and AsGR genes family [9,44], indicating that chromosomal polyploidization might be a driving factor in the formation and expansion of the AsAOX gene family. Typically, AOX genes feature a classic structure of four exons interrupted by three introns [23,41] (Figure 4C). Gene structural variations often arise in gene families through the gain or loss of introns during evolution [18]. Introns can influence gene recombination and alternative splicing of ncRNAs, thereby affecting gene evolution [47]. Among the 12 identified AsAOX genes in *A*. *sativa* (hexaploid), 4 AlAOX genes in *A*. *longiglumis* (diploid), and 9 AiAOX genes in *A*. *insularis* (tetraploid), most displayed the classical AOX gene structure. (Figure 4C). However, all AOX genes in Cluster 3 had multiple introns and exons, deviating from the classic structure, likely due to natural selection pressures [41,54]. Gene duplication and structural analysis revealed changes in the structure of some duplicated genes. For instance, *AsAOX7* and *AsAOX9* form a duplication pair but differ in the position and size of their first intron, suggesting that structural changes following gene duplication could drive the evolution of AsAOX genes [23] (Figure 4C and Figure 6). *AsAOX2* does not exhibit gene duplication events, and there are significant evolutionary divergences among AsAOX, AiAOX, and AlAOX genes. Moreover, *AsAOX2* performed exceptionally well in RT-qPCR analysis under five types of abiotic stresses in this study. Therefore, we infer that *AsAOX2* is likely a new gene generated through a mechanism different from gene duplication—specifically, gene fragment recombination [55]. Consequently, gene fragment recombination may have contributed to the expansion of the AsAOX gene family In summary, segmental duplication events, chromosomal polyploidization, structural variations following gene duplication and gene fragment recombination may all contribute to the expansion and evolution of the AsAOX gene family.

### 3.3. Exploration of Evolutionary Conservation and Divergence

AiAOXs and AlAOXs are considered ancestral genes to AsAOX genes [31]. Most of the genes exhibit strong evolutionary conservation, as reflected in the similar motif composition, gene structure, evolutionary branches, and orthologous gene pairs across multiple AOX genes (Figure 4 and Appendix A). Based on our findings, *AsAOX11*, *AsAOX5*, and *AsAOX12* are potential candidate genes with functions related to environmental stress resistance. Since *AiAOX4* and *AsAOX11* show a high degree of evolutionary conservation, it is likely that the potential functions of *AsAOX11* were inherited from its ancestral gene, *AiAOX4*. Similarly, we infer that the potential functions of *AsAOX5* and *AsAOX12* in resisting salt and drought stress likely originated from their respective ancestral genes, *AiAOX6* and *AiAOX3*. However, some AsAOX genes have diverged from their ancestral AiAOX genes in terms of motif composition or gene structure. We suggest that *AsAOX2*, *AsAOX4*, and *AsAOX8* may also have potential functions in stress resistance. However, *AsAOX2* shows differences in motif composition and gene structure compared to *AiAOX5*, *AiAOX6*, and *AiAOX7*, which are located on the same evolutionary branch, indicating functional divergence. Similarly, *AsAOX4* and *AsAOX8* have also diverged evolutionarily from several AiAOX genes on the same branch. Therefore, the potential stress-resistance functions of *AsAOX2*, *AsAOX4*, and *AsAOX8* may not have been inherited from their ancestral genes, but, rather, arose from gene fragment recombination during evolution [55]. Overall, there is both strong evolutionary conservation and divergence between the AsAOX genes and their ancestral AiAOX and AlAOX genes, contributing to the acquisition of potential functions by certain AsAOX genes.

As an important member of the *Poaceae* family, the evolutionary conservation of AsAOX genes in *Poaceae* has yet to be fully elucidated. Therefore, we explored the evolutionary relationships of the AOX gene family among *A. sativa* and three key *Poaceae* model species—*Oryza sativa*, *Triticum aestivum*, and *Brachypodium distachyon* (Appendix A). Our findings revealed both evolutionary conservation and divergence in AOX genes among *A. sativa* and these *Poaceae* model species. For example, *AsAOX1*, *AsAOX3*, and AsAOX6 share high similarity in motif composition and gene structure with OsAOX3, *TaAOX1–3*, and *BdAOX1*, which are located on the same evolutionary branch, indicating strong evolutionary conservation. However, other AsAOX, OsAOX, TaAOX, and BdAOX genes exhibit significant evolutionary divergence, characterized by differences in motif composition or gene structure. Thus, there is considerable evolutionary divergence and limited conservation of AOX genes among *A. sativa* and these three important Poaceae model species.

Additionally, we found that *AsAOX2* shows substantial evolutionary divergence not only within the 12 AsAOX genes but also when compared to AiAOX, AlAOX, OsAOX, TaAOX, and BdAOX genes. This pronounced evolutionary divergence may explain its superior performance in our RT-qPCR experiments under the five abiotic stress conditions tested in this study (Figure 9).

### 3.4. Preservation Mechanism of AsAOX Duplicated Genes

Duplication gene fates primarily include pseudogenization, neofunctionalization, and subfunctionalization [56,57,58,59]. Additionally, whole-genome duplication (WGD) is a major driving force in the evolution of eukaryotes [60]. It is reported that angiosperms have experienced multiple whole-genome duplication (WGD) events over the past 200 million years [61]. In this study, 9 out of 12 AsAOX members underwent gene duplication, suggesting that some AOX genes involved in WGD might have undergone pseudogenization and disappeared (Figure 6). In this study, a total of 16 AsAOX segmental duplication pairs, 7 AiAOX segmental duplication pairs, and 1 AlAOX segmental duplication pair were identified. Furthermore, AsAOX duplicated genes are responsive to various stresses and share the same domain; hence, all AsAOX duplicated genes have been preserved through subfunctionalization after duplication (Figure 6, Figure 9, and Figure 10 and Table 2).

### 3.5. Gene Expression and Tissue Specificity

RT-qPCR showed that *AsAOX12* notably responds to five stresses in both the leaf and root tissues of oat plants. Heat stress notably induced the expression of multiple AsAOX genes, especially *AsAOX2*, which exhibited high relative expression levels in both roots (30-fold CK) and leaves (210-fold CK). Additionally, *AsAOX2* shows high expression levels under other stresses. We also found that the motif composition, domain location, and gene structure of *AsAOX2* differ from those of other genes within the same cluster (Figure 4). Additionally, we found that *AsAOX7* responds notably to salt stress (Figure 10), despite containing very few ABREs (Figure 7). This response might be due to the regulation of *AsAOX7* by multiple transcription factors [18,23]. These transcription factors are not limited to the ABA-responsive pathway induced by salt stress but are part of a broader signaling network [62]. Previous research indicated that *TaAOX7*, *TaAOX9*, and *TaAOX10* of wheat expressions are downregulated under heat stress, and AsAOX4, being an orthologous gene, showed similar downregulation under heat stress [38] (Appendix A). *TaAOX16*, *TaAOX18*, and *TaAOX15* shared orthologous relationships with *AsAOX7*, *AsAOX9*, and *AsAOX10* (Cluster 3), all exhibiting no response to heat stress in leaves [38]. Tissue-specific analyses showed that *AsAOX11* was notably induced by five different abiotic stresses in leaves, while its expression was low in roots (Figure 9). In contrast, *AsAOX12* was downregulated in leaves under waterlogging and wounding stresses but upregulated in roots. Similarly, *TaAOX4* displayed high expression in leaves and low expression in roots, demonstrating tissue specificity. *TaAOX4*, an ortholog of *AsAOX11* and *AsAOX12*, also participates in TaAOX gene duplication events. [38] (Figure 9 and Appendix A). Their shared domains suggest that subfunctionalization following gene duplication might confer the tissue-specific expression in leaves and roots, aiding environmental adaptation [5] (Appendix A).

### 3.6. Gene Expression Regulation by CREs

CREs exist in the gene side sequence, a piece of DNA sequence that can affect gene expression, and their role is to participate in the regulation of gene expression. CREs are involved in gene transcriptional regulation and can predict potential gene functions based on their types and quantities within promoter regions [63,64]. Differences in AsAOX gene expression under various stresses might be attributed to the varying CRE types and quantities among AsAOX members [18]. Introns can have certain impacts on gene recombination and the alternative splicing of ncRNAs, thereby influencing gene evolution [47]. Previous studies showed that ABRE regulates gene expression under ABA-mediated stress [50]. Plants accumulate signaling molecules like ABA, activating mitochondrial retrograde regulation, inducing AOX gene expression, and providing stress resistance through various physiological effects [11,16]. ABA-enhanced AOX expression plays a crucial role in protecting photosystems and reducing ROS accumulation under drought stress [11,16]. The interaction between ABA regulation and AOX pathways suggests a certain level of interdependence [11,65]. Under ABA-mediated stress, PeAOX1a_1 (9 ABREs) exhibited the highest expression, whereas PeAOX1c, despite having the most ABREs (11), did not exhibit the highest expression [11]. In this study, *AsAOX8*, with the most ABREs (6), showed the highest expression under ABA-mediated stress in both roots and leaves in RT-qPCR analyses (Figure 9). However, *AsAOX2*, despite having fewer ABREs (3), responded notably to ABA-mediated stress. Furthermore, *AsAOX3*, an orthologous gene of PeAOX1a_1 with a substantial number of ABREs (5), did not respond notably to ABA-mediated stress (Appendix A and Figure 9). This suggests that gene response to ABA-mediated stress may be determined by a complex regulatory network involving multiple CREs rather than a single CRE [44,66].

### 3.7. Exploration of the Potential Functions of Some AsAOXs

We utilized RT-qPCR and CREs analysis to explore potential new functions of the AsAOX gene family under various abiotic stresses. RT-qPCR analysis revealed that *AsAOX2*, *AsAOX4*, and *AsAOX12* were notably expressed in response to wounding and waterlogging stresses. Additionally, *AsAOX8* and *AsAOX11* showed notable expression under Cr stress and waterlogging stress (Figure 9). Furthermore, *AsAOX2* and *AsAOX4* contain multiple W-boxs [50], suggesting that *AsAOX2* and *AsAOX4* may play roles in wounding stress resistance. For waterlogging and Cr stress, CREs analysis revealed that *AsAOX11* has the most ARE (2) [50], indicating that *AsAOX11* may have dual functions in resisting both waterlogging and Chromium stresses. *AsAOX8* possesses the highest number of ABREs (6) [50] and exhibits the highest expression levels under ABA-mediated stress (Figure 9), suggesting that *AsAOX8* may have a potential role in mitigating ABA-mediated stress. We found that the promoter of *AsAOX2* does not contain a specific heat stress-responsive CRE-HSE, yet it exhibits a strong response to heat stress (Figure 9) [67]. Previous studies have shown that high temperatures promote photorespiration, which in turn upregulates the AOX pathway, ultimately reducing cellular membrane damage [68]. Additionally, *AsAOX2* shows a significant response to heat stress in both root and leaf tissues (Figure 9). Therefore, *AsAOX2* may also have potential functions in mitigating heat stress. Interestingly, we observed that, under wounding stress, several duplicated genes in Cluster 1 displayed an intriguing pattern of expression (Figure 7A and Figure 9). Specifically, *AsAOX1*, *AsAOX11*, and *AsAOX12* exhibited significant downregulation, whereas *AsAOX3*, *AsAOX6*, and *AsAOX8* showed an overall upregulation. This finding aligns with CIPK6 homologous genes of cotton in previous research [69], implying that subfunctionalization may be responsible for altering the expression levels within Cluster 1. Furthermore, it is plausible that these genes are regulated positively or negatively by distinct mechanisms in response to wounding stress.

Previous studies have shown that overexpression of *PeAOX1b_2* from *Phyllostachys edulis* and *BanAOX1b* from *Brassica napus* enhances tolerance in transgenic plants under drought and salt stresses [11,19]. *AsAOX4*, *AsAOX5*, and *AsAOX2* are orthologous genes of *PeAOX1b_2*, while *AsAOX1*, *AsAOX3*, and *AsAOX6* are orthologous to *BanAOX1b* (Appendix A). Additionally, the promoters of *AsAOX2*, *AsAOX5*, and *AsAOX10* have the most DRE cores (2) [50] for responding to drought stress, and they share a similar domain (Appendix A). RNA-Seq data show that *AsAOX5* notably responds to drought stresses. Therefore, we infer that *AsAOX5* is most likely to have a potential function in resisting drought stresses. Regarding salt stress, *AsAOX12* responds notably to salt stress (Figure 9) and contains numerous ABREs (5) [70]. Therefore, *AsAOX12* likely plays a functional role in conferring resistance to salt stress.

### 3.8. Prospects for AsAOX Gene Family

In conclusion, our study identifies *AsAOX2*, 4, 5, 8, 11, and 12 as potential genes for resisting various abiotic stresses. Future research should validate these genes’ functions in *A. sativa* and other important crops to improve stress tolerance. Additionally, *AsAOX11* shows strong tissue specificity in roots and leaves under multiple stresses, highlighting the need to investigate its tissue-specific regulatory mechanisms. Furthermore, *AsAOX3*’s significant ABA response despite having few ABREs suggests it is regulated by a complex CREs network. Future studies should explore these regulatory networks to better understand *AsAOX3*’s role in ABA-mediated stress responses.

## 4. Materials and Methods

### 4.1. Identification and Classification of AsAOXs Gene Family in A. sativa (AsAOXs), A. longiglumis (AlAOXs), and A. insularis (AiAOXs)

The various files required for oat (*A. sativa*, cv. Sang, hexaploid) and two its potential ancestral species, *A. longiglumis* (cv. CN58138, Diploid) and *A. insularis* (cv. BYU209, tetraploid), were obtained from the GrainGenes database (https://wheat.pw.usda.gov/GG3/, accessed on 16 October 2023) [71]. Based on TBtools-II v2.119, a BLASTP analysis was performed using the whole proteome files of *A. sativa*, *A. longiglumis*, and *A. insularis*, along with the AOX protein sequence of a model plant [72]. The Hidden Markov Model (HMM) matrix file from the Pfam database (http://pfam-legacy.xfam.org/, accessed on 16 October 2023) with Pfam ID number PF01786 was utilized for HMM Search using TBtools-II v2.119 software. Candidates identified as potentially containing the AOX gene family domain were selected based on the consensus results derived from both BLASTP and HMM Search. The protein sequences of these candidates were uploaded to the ‘Batch CD-Search pattern’ of NCBI (https://www.ncbi.nlm.nih.gov/Structure/bwrpsb/bwrpsb.cgi, accessed on 16 October 2023) [73], facilitating the identification of AsAOXs, AlAOXs, and AiAOXs gene family members with complete domains.

### 4.2. Analysis of Conserved Motif, Structural Domain, and Gene Structure of AsAOXs, AlAOXs, and AiAOXs Gene Family

To identify conserved motifs, gene structures, and domains within the AsAOX, AlAOX, and AiAOX gene families, the protein sequences of the identified members were uploaded to the MEME platform (https://meme-suite.org/meme/tools/meme, accessed on 16 October 2023) [74]. The identified conserved motifs, gene structures, and domains were subsequently visualized using the ‘Visualize MEME MAST Motif Pattern’, ‘Visualize Gene Structure’, and ‘Visualize NCBI CDD Domain Pattern’ tools available in the TBtools-II v2.119 software.

### 4.3. Gene Location, Protein Characteristics, and Subcellular Localization Prediction of AsAOXs, AlAOXs, and AiAOXs Gene Family

The ‘Gene Location visualization from GTF/GFF pattern’ function in the TBtools-II v2.119 software was used to visualize the chromosomal localization of each member of the AsAOX, AiAOX, and AlAOX gene family. Additionally, the ‘Protein Parameter Calculation’ tool in TBtools-II v2.119 was employed to analyze protein characteristics for the protein sequences of the AsAOXs, AiAOXs, and AlAOXs gene family members. Furthermore, WoLF PSORT [75] (https://wolfpsort.hgc.jp/, accessed on 28 October 2023) was utilized to predict the subcellular localization of AsAOXs, AlAOXs, and AiAOXs proteins.

### 4.4. Analysis of Gene Duplication, Selection Pressure, and Gene Synteny of AOXs Genes

A Gene Synteny analysis was performed in *A. sativa*, *A. longiglumis*, *A. insularis*, *Triticum aestivum*, *Hordeum vulgare*, *Secale cereale*, *Oryza sativa*, *Sorghum bicolor*, and *Brachypodium distachyon*. The ‘One Step MCScanX-Super Fast’ plugin in TBtools-II v2.119 was employed for synteny analysis and gene duplication analysis. Finally, the ‘Multiple Synteny Plot Pattern’ was used to visualize gene synteny. The ‘Simple Ka/Ks Calculator (NG) pattern’ in TBtools-II v2.119 was utilized to calculate substitution rates (Ks), nonsynonymous substitution rates (Ka), and their ratio (Ka/Ks). Finally, the OrthoVenn3 tool (https://orthovenn3.bioinfotoolkits.net/start/db, accessed on 3 May 2024) was used to infer orthologous gene pairs between AsAOXs and AOXs in other plants [76].

### 4.5. Identification of Cis-Regulatory Elements in Promoter Region of AsAOXs Gene Family

Studying the cis-regulatory elements in the gene promoter region facilitates the exploration of the functional and adaptive mechanisms of individual gene family members in diverse environments. Therefore, in order to analyze the quantity and functionality of cis-regulatory elements (CREs) in the AsAOXs, AiAOXs, and AlAOXs gene families, we utilized the ‘GTF/GFF3 Sequences Extract pattern’ of TBtools-II v2.119 to extract the upstream 1500 bp sequences of the AsAOXs genes as promoter regions. Subsequently, we uploaded these sequences to PlantCARE (https://bioinformatics.psb.ugent.be/webtools/plantcare/html/, accessed on 21 October 2023) for cis-acting element identification, followed by visualization using an R script and Hiplot website (https://hiplot.com.cn/cloud-tool/drawing-tool/list, accessed on 21 October 2023) [77].

### 4.6. Multi-Sequence Alignment and Phylogenetic Analysis of AOXs Gene Family

Phylogenetic analysis provides valuable insights into the evolutionary relationships among different species or the genetic relationships within the same species. The protein sequences of each species were used for multiple sequence alignment using the MUSCLE algorithm in MEGA v7.0 software [78,79]. Subsequently, the maximum-likelihood (ML) method was employed to construct ML phylogenetic trees for AOXs. To enhance the reliability of the phylogenetic trees, both trees were built with 1000 bootstrap replicates. Finally, the obtained phylogenetic tree files in nwk format were beautified and visualized using EvolView [80], an online platform (https://www.evolgenius.info/evolview-v2/#login, accessed on 6 December 2023).

### 4.7. Prediction of Protein 3D Structure and Protein–Protein Interaction

The function of proteins is determined by their three-dimensional structure, making the study of protein structures crucial for understanding the functions of genes. In this study, the protein sequences of 12 AsAOXs genes were uploaded to the online website SWISS-MODEL (https://swissmodel.expasy.org/, accessed on 21 December 2023) [81]. Subsequently, a homology search was performed within the template PDB database [82] (https://www.rcsb.org/, accessed on 21 December 2023) of SWISS-MODEL to identify known species with high Sequence Identity, Sequence Similarity, and GMQE Value to the AsAOXs gene sequences. Finally, the protein structures were visualized and refined using the protein structure visualization software VMD [83]. Additionally, we referred to methods from previous studies to construct the protein–protein interaction (PPI) network [9]. We utilized the STRING database (https://cn.string-db.org/, accessed on 21 December 2023) to predict the interactions among AsAOXs proteins, as well as their interactions with other proteins, and the results were visualized and beautified using Cytoscape [84].

### 4.8. Analysis of Gene Ontology (GO) Enrichment of AOXs Genes in A. sativa

GO enrichment analysis is helpful for studying plant environmental adaptation and stress responses. Therefore, we uploaded the protein sequence file of the AsAOXs to EGGNOG-MAPPER (http://eggnog-mapper.embl.de/, accessed on 21 December 2023) for protein function annotation [85]. Then, we performed GO enrichment analysis using the ‘eggNOG-mapper pattern’ and ‘GO Enrichment pattern’ in TBtools-II v2.119. Finally, we visualized the results of the GO enrichment analysis using the Bioinformatics online platform (https://www.bioinformatics.com.cn/, accessed on 14 March 2024) [86].

### 4.9. Information, Culture Environment, and Treatment of Oat Plant Material

In this study, *A. sativa* (cv. Baylor) was used, and the seeds were germinated in quartz sand stored in a greenhouse. In a greenhouse with a photosynthetically active radiation of 200–240 μmol m^−2^ s^−1^, a temperature regime of 25 °C during the day and 15 °C at night, and a photoperiod of 16 h light and 8 h dark, oat seeds were sterilized with a 2% sodium hypochlorite solution for 2 min. To prevent residual sodium hypochlorite from affecting seed germination, the seeds were rinsed three times with distilled water. They were then germinated on quartz sand under white fluorescent light at 70% relative humidity. To ensure consistent growth conditions, the germinated seedlings were transferred to Hoagland nutrient solution for cultivation after seven days. To prevent low oxygen levels in the nutrient solution from affecting the experimental results, we aerate the solution using a pump continuously. Additionally, given the continuous consumption of nutrients in the solution, we replaced the nutrient solution once daily. After two weeks, the oat seedlings were placed under five different abiotic stress treatments. Root and leaf tissues of the oat seedlings were sampled at 0 h, 3 h, 6 h, 9 h, 24 h, and 48 h and stored at −80 °C for further analysis. Additionally, we used non-stressed oat seedlings as controls to minimize the influence of other factors on the experimental results. Waterlogging stress: The nutrient solution was adjusted to maintain a 5 cm depth above the roots, simulating waterlogging conditions. To minimize the impact of O_2_ levels in the nutrient solution on waterlogging stress, we replaced the nutrient solution once daily and did not use any aeration measures. Abscisic acid (ABA) stress: Oat seedlings were uniformly sprayed with 100 μM ABA (0.1% Tween-20) twice daily, in the morning and evening, to induce ABA-mediated stress responses. Additionally, to minimize the impact of ABA leaching into the nutrient solution on the experimental results, we placed a foam board clamp above the nutrient solution but below the oat seedling leaves. Heat stress: to simulate heat conditions, oat plants were subjected to a temperature regime of 40 °C during the day and 30 °C during the night. Wounding stress: oat leaves were uniformly trimmed to the same height to simulate wounding stress. Heavy metal stress: a 200 μM potassium chromate solution was dissolved in the nutrient solution to induce heavy metal stress.

### 4.10. RT-qPCR and RNA-Seq Analysis of Oat

The total RNA extraction, reverse transcription, and RT-qPCR analysis of oat samples were performed using the following reagents: Direct-zol™ RNA MiniPrep Kit (Zymo Research, Beijing, China) for RNA extraction, ABScript III RT Master Mix for qPCR with Gdna Remover (Abclonal, Wuhan, China) for reverse transcription, and Genious 2× SYBR Green Fast Qpcr Mix (Abclonal, Wuhan, China) for RT-qPCR. The RT-qPCR reactions were conducted on the CXF96 Connect™ Real-Time System (Bio-Rad, Singapore). To calculate the relative expression levels of the 12 identified AsAOXs gene members under five abiotic stress conditions (using the 2^−ΔΔCt^ method) [87], the *AsEIF4A* gene was selected as the reference gene for stable expression [88]. We downloaded transcriptome data for salt stress (SRR5097544–SRR5097549) [89] and drought stress (SRR10718331–SRR10718339) in oat from the NCBI website (https://www.ncbi.nlm.nih.gov/protein, accessed on 16 March 2024). Finally, we obtained gene expression levels for 12 AsAOX genes under salt stress and drought stress.

## 5. Conclusions

This study identified the AOX gene family members in *A. sativa*, *A. insularis*, and *A. longiglumis*. Most of the AsAOX, AiAOX, and AlAOX gene family members show strong evolutionary conservation. Additionally, the AOX gene family exhibits strong evolutionary conservation and weak evolutionary divergence in these three *Avena* species. However, in the four important *Poaceae* plants, it shows weak evolutionary conservation and strong evolutionary divergence. The evolution and expansion of the AsAOX gene family may have been driven by segmental duplications, chromosome polyploidization, gene structural variations, and gene fragment recombination. Some AsAOX genes, such as *AsAOX11*, exhibit notable tissue specificity between leaves and roots. Additionally, *AsAOX2*, 4, 5, 8, 11, and 12 may have potential functions in resisting various abiotic stresses.

## Figures and Tables

**Figure 1 ijms-25-09383-f001:**
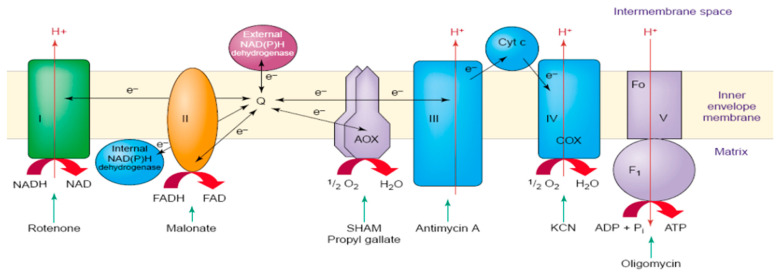
Schematic diagram of the mitochondrial respiratory electron transport chain in higher plants [13]. It illustrates the position of complexes I to V and AOX in the electron transport chain.

**Figure 2 ijms-25-09383-f002:**
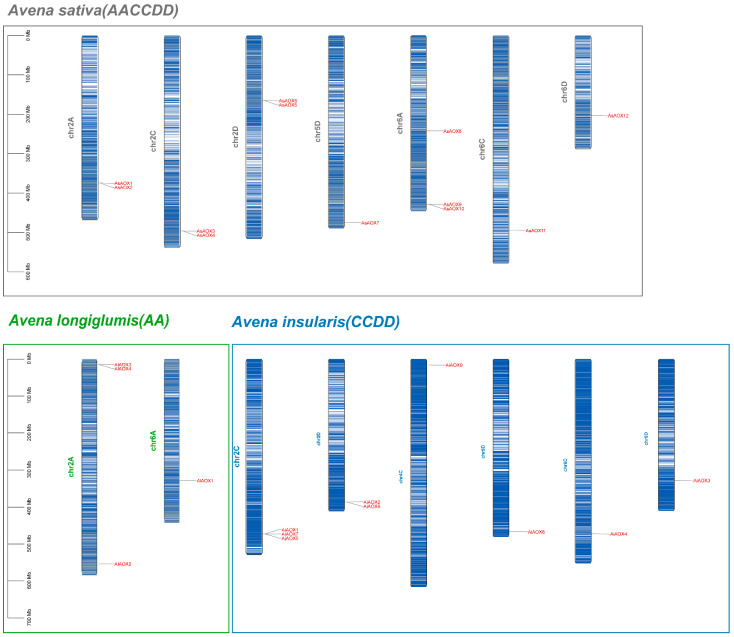
The chromosomal localization of AsAOXs, AlAOXs, and AiAOXs was depicted using brown, green, and blue colors to represent *A. sativa*, *A. longiglumis*, and *A. insularis*, respectively. The shades of these colors on the chromosomes reflect varying gene densities.

**Figure 3 ijms-25-09383-f003:**
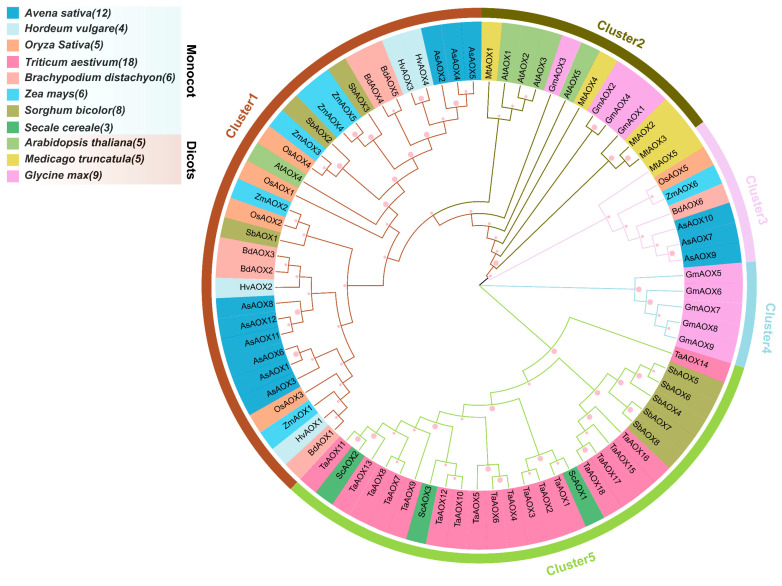
An ML phylogenetic tree of 81 AOX protein sequences from 11 species (*A. sativa*, *Oryza sativa*, *Arabidopsis thaliana*, *Triticum aestivum*, *Zea mays*, *Brachypodium distachyon*, *Hordeum vulgare*, *Glycine max*, *Medicago truncatula*, *Sorghum bicolor*, *Secale cereale*) was constructed using MEGA v7.0. Pink circles of different sizes represent various bootstrap values, with the circle size indicating bootstrap values ranging from 0 to 100. The number following each species’ Latin name indicates the quantity of AOX members from that species.

**Figure 4 ijms-25-09383-f004:**
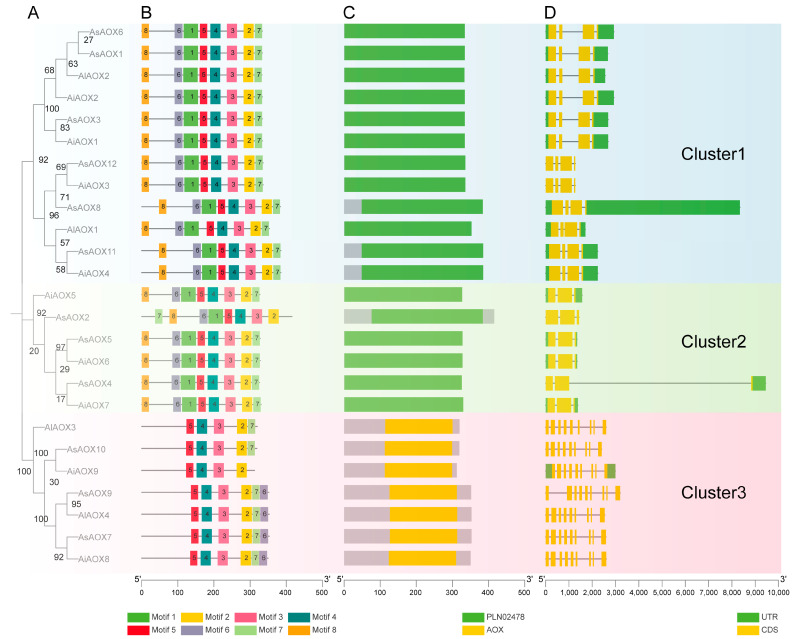
(**A**) Phylogenetic analysis of 25 AOX protein sequences (12 AsAOX, 9 AiAOX, 4 AlAOX), with bootstrap values depicted on branches ranging from 0 to 100. (**B**) Conservation motif analysis. (**C**) Conservation domain analysis. (**D**) Gene structure composition, where black lines, yellow boxes, and green boxes represent introns, exons (CDS), and UTRs (Untranslated Regions), respectively. Different colored backgrounds indicate clustering of different AOX gene members.

**Figure 5 ijms-25-09383-f005:**
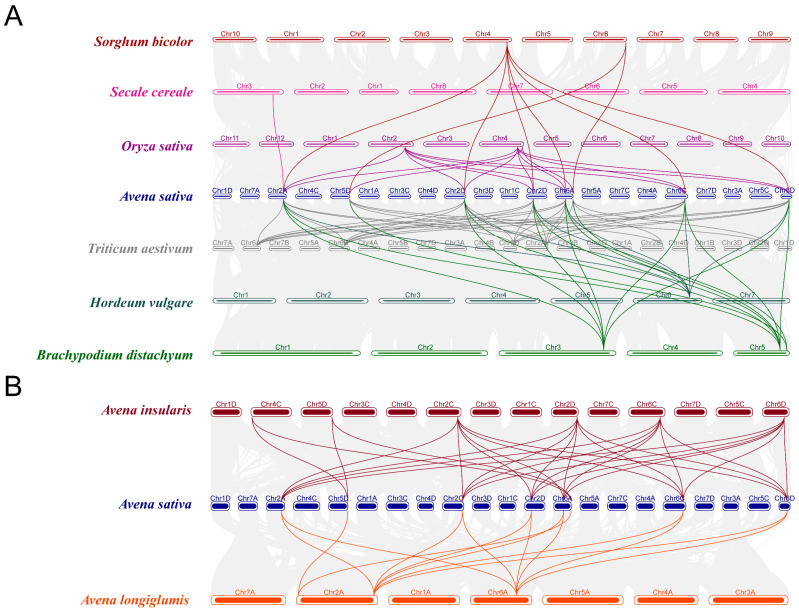
(**A**) Synteny analysis of AsAOXs with the AOX gene families of eight species. These include *A. sativa*, closely related Triticeae cereals *Triticum aestivum*, *Hordeum vulgare*, and *Secale cereale*, globally important crops Oryza sativa and *Sorghum bicolor*, as well as the model plant *Brachypodium distachyon*. (**B**) Syteny analysis of *A. sativa* with its possible two ancestral species *A. insularis* and *A. longiglumis*. Different colored lines represent synteny gene pairs between the AOX genes of the respective species and AsAOX genes, while grey lines indicate synteny blocks in other genes.

**Figure 6 ijms-25-09383-f006:**
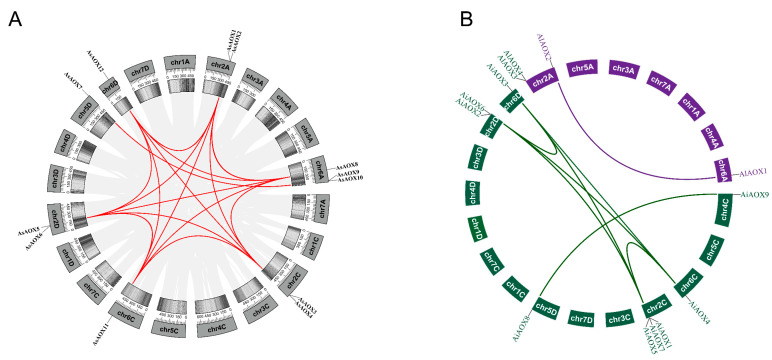
Gene duplication analysis of AsAOXs, AiAOXs, and AlAOXs. (**A**) Gene duplication analysis of AsAOXs. Red lines represent paralogous gene pairs of the AsAOX gene family. Shades of grey indicate the density of genes. (**B**) The green and purple lines represent paralogous gene pairs of the AiAOX and AlAOX gene families, respectively.

**Figure 7 ijms-25-09383-f007:**
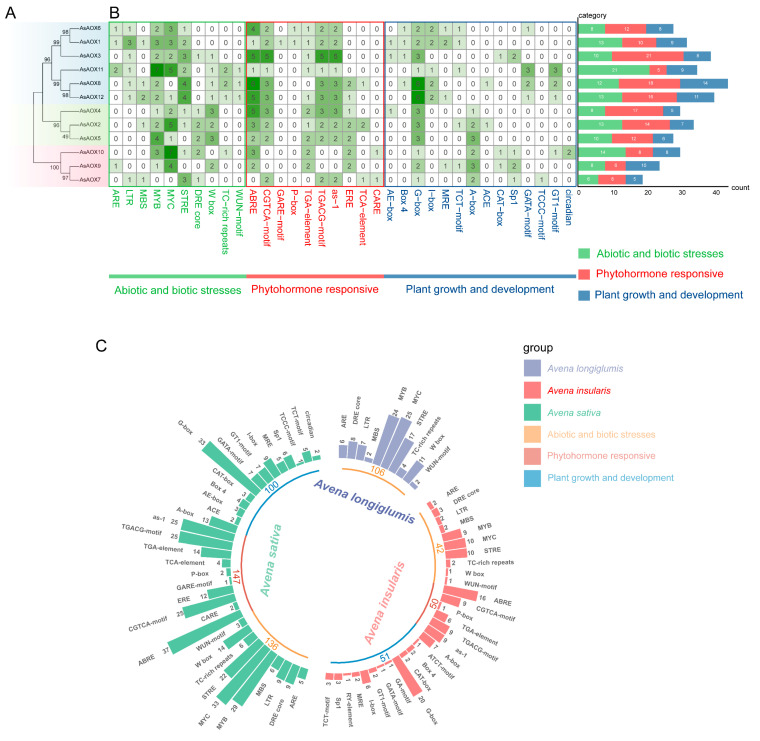
AsAOXs, AiAOXs, and AlAOXs gene promoter region CREs analysis. (**A**) An ML phylogenetic tree of AsAOXs, with bootstrap values indicated on branches ranging from 0 to 100. (**B**) Analysis of the number, distribution, and classification of Cis-regulatory elements in AsAOXs. (**C**) Comparative analysis of CREs in the promoter regions of AsAOXs, AlAOXs, and AiAOXs genes.

**Figure 8 ijms-25-09383-f008:**
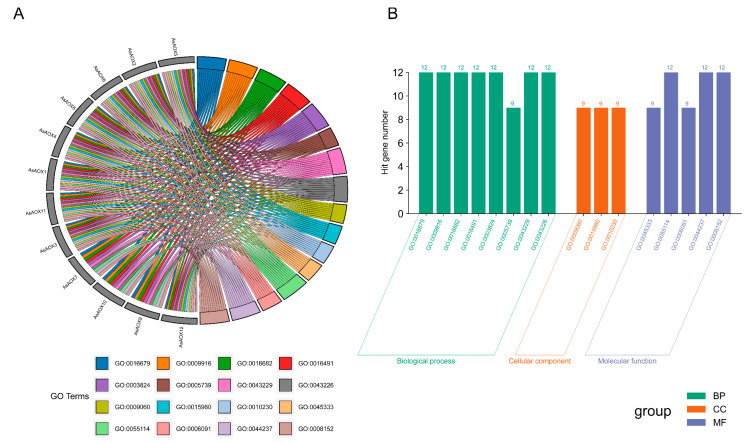
GO enrichment analysis of 12 AsAOX genes. (**A**) GO terms enriched for each AsAOX gene. (**B**) Number of AsAOX genes contained within each GO term.

**Figure 9 ijms-25-09383-f009:**
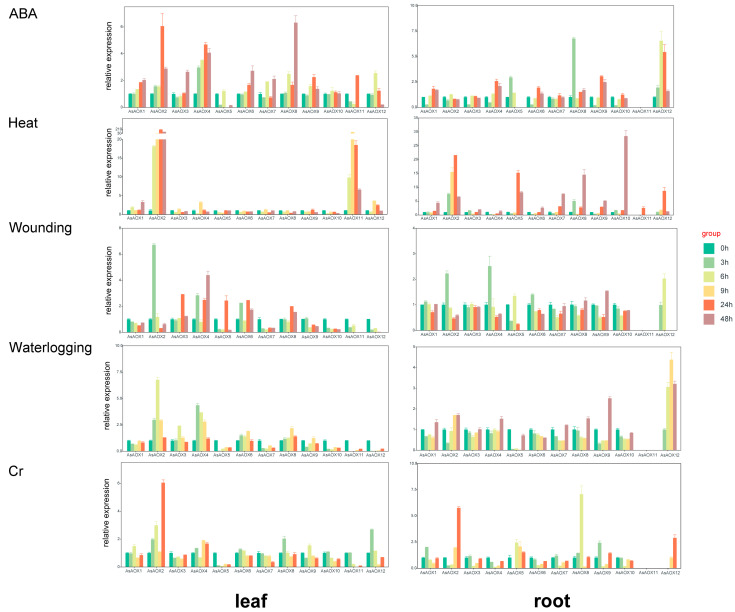
RT-qPCR analysis reveals the relative expression levels of 12 AsAOX genes at different time points (0 h, 3 h, 6 h, 9 h, 24 h, 48 h) under five abiotic stress conditions in oat leaves and roots. If expression levels at 0 h are too low to detect, the subsequent time point is used as control (ck). Error bars represent the standard error of three biological replicates.

**Figure 10 ijms-25-09383-f010:**
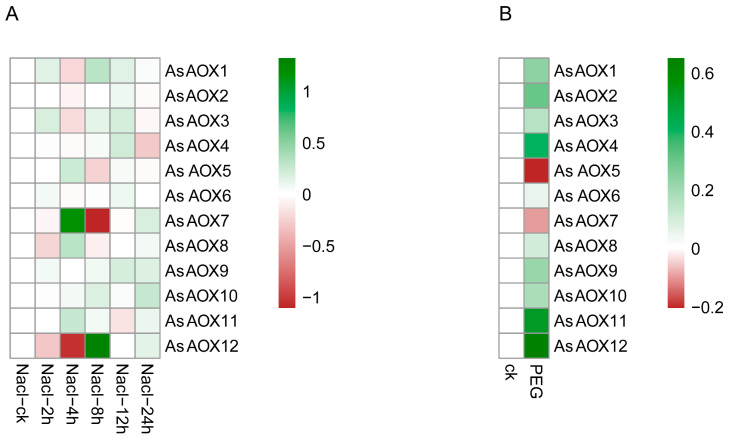
(**A**,**B**) depict the expression profiles of 12 AsAOX genes under drought stress and salt stress from RNA-Seq data, respectively. Shades of red and green indicate the degree of downregulation and upregulation, respectively.

**Table 1 ijms-25-09383-t001:** The physicochemical properties and subcellular location prediction of AsAOX, AiAOX, and AlAOX.

Gene Name	AA	MW	PI	II	AI	GRAVY	SP
*AsAOX1*	334	37,413.71	8.39	42.64	77.78	−0.237	Mitochondrion
*AsAOX2*	416	46,772.25	8.88	55.29	77.26	−0.344	Nucleus/Mitochondrion
*AsAOX3*	334	37,384.71	8.68	40.08	77.78	−0.236	Mitochondrion
*AsAOX4*	326	36,735.96	6.96	47.5	80.61	−0.24	Chloroplast
*AsAOX5*	328	36,800.07	7.3	44.86	82.53	−0.21	Chloroplast
*AsAOX6*	334	37,397.71	8.39	42.64	78.08	−0.23	Mitochondrion
*AsAOX7*	353	40,196.47	5.18	68.02	73.03	−0.305	Chloroplast
*AsAOX8*	384	42,425.61	9.28	38.98	86.69	−0.131	Chloroplast
*AsAOX9*	352	40,143.26	5.01	61.49	71.31	−0.339	Chloroplast
*AsAOX10*	319	36,735.62	5.47	61.89	76.52	−0.251	Mitochondrion
*AsAOX11*	385	42,535.73	9.3	39.05	86.75	−0.135	Chloroplast
*AsAOX12*	336	37,379.95	7.83	32.25	88.33	−0.115	Chloroplast
*AiAOX1*	334	37,384.71	8.68	40.08	77.78	−0.236	Mitochondrion
*AiAOX2*	334	37,427.74	8.39	42.64	77.78	−0.237	Mitochondrion
*AiAOX3*	336	37,335.94	8.4	32.5	88.63	−0.099	Chloroplast
*AiAOX4*	385	42,550.74	9.39	39.83	86.49	−0.141	Mitochondrion
*AiAOX5*	327	36,770.99	6.96	44.56	82.45	−0.233	Chloroplast
*AiAOX6*	328	36,800.07	7.3	44.86	82.53	−0.21	Chloroplast
*AiAOX7*	330	37,123.33	6.74	44.06	79.09	−0.25	Chloroplast
*AiAOX8*	350	40,088.34	5.24	65.94	70.57	−0.346	Chloroplast
*AiAOX9*	312	36,018.92	5.69	63.06	76.67	−0.239	Chloroplast
*AlAOX1*	353	39,421.42	8.84	32.72	88.78	−0.102	Chloroplast
*AlAOX2*	333	37,268.59	8.69	43.48	78.02	−0.226	Mitochondrion
*AlAOX3*	320	36,567.26	5.27	58.71	76.28	−0.221	Chloroplast
*AlAOX4*	353	40,197.49	4.96	62.99	73.03	−0.295	Chloroplast

**Table 2 ijms-25-09383-t002:** The paralogous gene pairs of AsAOX gene family, AiAOX gene family, and AlAOX gene family.

Seq1	Seq2	Ka	Ks	Ka/Ks	Duplication Type
*AsAOX1*	*AsAOX3*	0.006618	0.101104	0.065459	Segmental
*AsAOX1*	*AsAOX6*	0.00132	0.105669	0.012488	Segmental
*AsAOX1*	*AsAOX8*	0.130136	0.530657	0.245235	Segmental
*AsAOX1*	*AsAOX11*	0.133032	0.510456	0.260614	Segmental
*AsAOX1*	*AsAOX12*	0.130136	0.522285	0.249166	Segmental
*AsAOX3*	*AsAOX6*	0.005291	0.139623	0.037895	Segmental
*AsAOX4*	*AsAOX8*	0.252513	0.52896	0.477377	Segmental
*AsAOX3*	*AsAOX11*	0.140535	0.50901	0.276094	Segmental
*AsAOX3*	*AsAOX12*	0.137656	0.52901	0.260215	Segmental
*AsAOX6*	*AsAOX8*	0.129355	0.595988	0.217042	Segmental
*AsAOX6*	*AsAOX11*	0.132251	0.582591	0.227004	Segmental
*AsAOX6*	*AsAOX12*	0.129355	0.595988	0.217042	Segmental
*AsAOX7*	*AsAOX9*	0.033025	0.237917	0.138808	Segmental
*AsAOX8*	*AsAOX11*	0.015335	0.097405	0.157438	Segmental
*AsAOX8*	*AsAOX12*	0.001329	0.052867	0.025131	Segmental
*AsAOX11*	*AsAOX12*	0.006671	0.082786	0.080584	Segmental
*AiAOX1*	*AiAOX2*	0.00662	0.139623	0.047411	Segmental
*AiAOX1*	*AiAOX4*	0.142912	0.514069	0.278002	Segmental
*AiAOX1*	*AiAOX3*	0.139355	0.527957	0.263951	Segmental
*AiAOX2*	*AiAOX4*	0.139859	0.587538	0.238042	Segmental
*AiAOX2*	*AiAOX3*	0.131048	0.594732	0.220347	Segmental
*AiAOX9*	*AiAOX8*	0.088943	0.318747	0.27904	Segmental
*AiAOX4*	*AiAOX3*	0.010707	0.087054	0.122996	Segmental
*AlAOX2*	*AlAOX1*	0.13146	0.49955	0.263157	Segmental

## Data Availability

Data are contained within the article and Appendix A.

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
