# Peer review of "Exploring Evolutionary Pathways and Abiotic Stress Responses through Genome-Wide Identification and Analysis of the Alternative Oxidase (AOX) Gene Family in Common Oat (Avena sativa)"

_ijms, 2024, doi:10.3390/ijms25179383_

Round 1
Reviewer 1 Report
Comments and Suggestions for Authors
1. Line 52: ETC should be spelled out as this is the first occurrence. Please consider a diagram outlining this. Some people may not be familiar/ remember complexes III and IV.
2. Line 85: suggest "ABA-mediated stress", as ABA alone isn't a stressor in the conventional sense. The authors switch from mentioning Cr in lines 38 and 82 to cadmium (Cd) in line 85. Was this intentional?
3. Section 2.9: an image of the plants at this stage/ the experimental setup would be preferable. Was the nutrient solution aerated? What was the photoperiod, temperature, and light level? Did the authors ascertain O2 levels in the water for the waterlogging treatment (hypoxia versus anoxia)? Was any adjuvent added to the ABA solution to ensure solution uptake? Assuming that the plants were still in nutrient solution and that the ABA spray would run off into the nutrient solution, are the authors concerned about foliar- versus root-sourced ABA effects? Please use PPFD units (mmole photons M-2 s-1). Measurement in lux is inappropriate for plant studies. If the plants were still in nutrient solution, what was the temperature of the nutrient solution? Could the nutrient solution have buffered the temperature experienced by the plants? Did the authors consider that K2CrO4 solution can impart an osmotic stress on the plants depending on the final volume? What was the final concentration of the K2CrO4? Would the resulting osmotic stress confound any effects of the Cr stress? How would the authors account for such a stress?
4. Section 2.10: current best practice is two or more endogenous reference genes. Only AsEIF4A was selected, based on Yang et al. (2020; ref 56), however, those authors demonstrated that AsEIF4A and Heterogeneous nuclear ribonucleoprotein 27C (HNR) were best in tandem overall, and that "UBC21, an example of a four-copy duplicated RG, along with HNR was identified as the most stable RG set in shoots and roots of oat seedlings." The obvious question is why didn't the authors of the current study follow suit? Moreover, did the authors of the current study verify the stability of AsEIF4A in the tissues and conditions used? In addition, Yang et al. (2020; ref 56) used shoots (presumably including leaves). Do the authors mean shoots plus leaves or leaves only? What are the potential consequences of any difference? The authors need to include the citations from which these publicly available data ((SRR5097544–SRR5097549) and drought stress (SRR10718331–SRR10718339)) are taken or include the methods in a Supplementary file. Are the methods comparable to those presented in the current Materials and Methods? Were the plants comparable to those used in the current study? For example, young plant responses may not equate to older plant responses. Plant organ structure and developmental processes may influence how a plant responds to stimuli.
5. Lines 247-248: why is this important?
6. Lines 454-456: please confirm 210 fold versus 210 times. They are mathematically distinct resulting in drastically different values. Unfortunately, they are conflated in analyses such as this.
7. Section 4.4: the authors used 0 time samples as the control unless the signal was undetectable, in which case the subsequent sample was used as the control. A question arises as to circadian rhythms and whether any of the AOX genes have natural fluctuations that may be part of any increases or decreases seen. When during the photoperiod were the treatments initiated?
8. Line 598: was CRE defined before? If not, it should be defined here.
9. Line 614: "significantly" in the statistical sense or a qualitative sense?
10. Line 630: it is understood that the authors use "ABA stress" as a shorthand for ABA-mediated responses, but it is jarring, nonetheless.
11. Line 634: perhaps "stimulates" or "upregulates" rather than "enhances".
Author Response
|
3. Point-by-point response to Comments and Suggestions for Authors Comment 1: Line 52: ETC should be spelled out as this is the first occurrence. Response: Thank you for your careful review. It is indeed necessary to spell out the full term upon first mention. We have replaced the first occurrence of "ETC" in the manuscript with "Electron Transfer Chain (ETC)." Line51-52: Replace “ETC“ with “Electron Transfer Chain (ETC) (Figure 1)“
Comment 2: Please consider a diagram outlining this. Some people may not be familiar/remember complexes III and IV. Response: Thank you for your careful review. We have added a schematic diagram of the ETC to the manuscript to help you and readers better understand complexes III, IV, and AOX. Line101-103: Add: “Figure 1: Schematic diagram of the mitochondrial respiratory electron transport chain in higher plants[1]. It illustrates the positions of complexes I to V and AOX with the electron transport chain.”
Comment 3: Line 85: Suggest "ABA-mediated stress," as ABA alone isn't a stressor in the conventional sense. Response: Thank you for the suggestion. We have replaced "ABA stress" with the more accurate term "ABA-mediated stress" in the manuscript. Line29: Replace “ABA stress” with “ABA-mediated stress” Line361: Replace “ABA stress” with “ABA-mediated stress” Line550: Replace “ABA stress” with “ABA-mediated stress” Line556: Replace “ABA stress” with “ABA-mediated stress” Line559: Replace “ABA stress” with “ABA-mediated stress” Line561: Replace “ABA stress” with “ABA-mediated stress” Line564: Replace “ABA stress” with “ABA-mediated stress” Line582: Replace “ABA stress” with “ABA-mediated stress” Line583-584: Replace “ABA stress” with “ABA-mediated stress” Line859: Replace “ABA stress” with “ABA-mediated stress”
Comment 4: The authors switch from mentioning Cr in lines 38 and 82 to cadmium (Cd) in line 85. Was this intentional? Response: We apologize for the oversight Cadmium (Cd) was mistakenly mentioned in the manuscript, and we have corrected this error. Line94: Replace “Cadmium (Cd) stress” with “Cr stress”
Comment 5: Section 2.9: An image of the plants at this stage/the experimental setup would be preferable. Response: Thank you for your suggestion. We apologize for not providing a detailed experimental design in the Materials and Methods section, which may have caused confusion. Due to time constraints, we may not be able to provide an illustration of the experimental setup at this stage, but we have enhanced the description in the Materials and Methods section for your review.
Comment 6: Was the nutrient solution aerated? Response: Thank you for the question. As seedlings continuously absorb oxygen from the nutrient solution, leading to anaerobic stress, we aerated the nutrient solution continuously using a pump and replaced solution once daily to minimize any effects of anaerobic stress on the experiment. Sorry for the oversight in not providing a detailed description in the Materials and Methods section. We have now added the relevant details for your review. Line838-840: Add: “To prevent low oxygen levels in the nutrient solution from affecting the experimental results, we continuously aerate the solution using a pump. Additionally, given the continuous consumption of nutrients in the solution, we replace the nutrient solution once daily."
Comment 7: What was the photoperiod, temperature, and light level? Response: Thank you for the question. We apologize for not providing this information earlier. The photoperiod in the greenhouse was 16 hours of daylight and 8 hours of darkness, with a temperature cycle of 25°C during the day and 15°C at night, and a light level of 200-240 μmol m−2 s−1. Sorry for the oversight in not providing a detailed description in the Materials and Methods section. We have now added the relevant details for your review. Line831-834: Add: “In a greenhouse with a photosynthetically active radiation of 200-240 μmol m−2 s−1, a temperature regime of 25°C during the day and 15°C at night, and a photoperiod of 16 hours light and 8 hours dark, oat seeds were sterilized with a 2% sodium hypochlorite solution for 2 minutes.”
Comment 8: Did the authors ascertain O2 levels in the water for the waterlogging treatment (hypoxia versus anoxia)? Response: Thank you for your question. We reviewed the papers about waterlogging stress before conducting our experiment and referred to their methods. However, these studies did not specify whether they measured the oxygen content in the nutrient solution in real-time during waterlogging stress [2-14]. The oxygen content in the nutrient solution indeed slightly affects the results of waterlogging stress studies. Unfortunately, due to the lack of equipment, we were unable to measure the oxygen content in real-time. However, to ensure a constant and consistent low-oxygen condition in the nutrient solution, we replaced the nutrient solution once daily and did not use any aeration measures. Oat seedlings were indeed subjected to waterlogging stress. Your suggestion would certainly enhance the accuracy of the results. We will take your suggestions into account in future similar studies. Sorry for the oversight in not providing a detailed description in the Materials and Methods section. We have now added the relevant details for your review. Line855-867: Add: “To minimize the impact of O2 levels in the nutrient solution on waterlogging stress, we replaced the nutrient solution once daily and did not use any aeration measures.”
Comment 9: Was any adjuvant added to the ABA solution to ensure solution uptake? Response: Thank you for your question. Indeed, applying ABA solution alone to leaves would result in poor absorption. Therefore, we added 0.1% Tween-20 to the ABA solution to promote uniform distribution and adhesion on the leaf surface, enhancing the absorption of ABA by oat seedlings. We have added the relevant descriptions to the Materials and Methods section for your review. Sorry for the oversight in not providing a detailed description in the Materials and Methods section. We have now added the relevant details for your review. Line858-859: Add: “(0.1% Tween-20) twice daily, in the morning and evening”
Comment 10: Assuming that the plants were still in nutrient solution and that the ABA spray would run off into the nutrient solution, are the authors concerned about foliar- versus root-sourced ABA effects? Response: Thank you for the question. To avoid interference with the experiment, we used foam boards to prevent the ABA solution from flowing into the nutrient solution during foliar application. Sorry for the oversight in not providing a detailed description in the Materials and Methods section. We have now added the relevant details for your review. Line859-861: Add: “Additionally, to minimize the impact of ABA leaching into the nutrient solution on the experimental results, we placed a foam board clamp above the nutrient solution but below the oat seedling leaves.”
Comment 11: Please use PPFD units (mmole photons m−2 s−1). Measurement in lux is inappropriate for plant studies. Response: Thank you for pointing out this issue. We have corrected the units in the manuscript to reflect proper plant study standards. Line831-832: Add: “photosynthetically active radiation of 200-240 μmol m−2 s−1”
Comment 12: If the plants were still in nutrient solution, what was the temperature of the nutrient solution? Could the nutrient solution have buffered the temperature experienced by the plants? Response: Thank you for your question. We used a temperature of 40°C to induce heat stress, and since the nutrient solution has a specific heat capacity, it did buffer the temperature experienced by the plant roots, with the maximum temperature measured in the solution being 35°C (heat stress[15]). Thus, both the roots and leaves were exposed to heat stress, though the roots experienced slightly less stress than the leaves. We believe that this buffering effect did not significantly affect the experimental results. However, we acknowledge that this factor may have contributed to the lower expression levels in the roots compared to the leaves under heat stress. We will consider this suggestion in the future study.
Comment 13: Did the authors consider that K2CrO4 solution can impart an osmotic stress on the plants depending on the final volume? What was the final concentration of the K2CrO4? Would the resulting osmotic stress confound any effects of the Cr stress? How would the authors account for such stress? Response: Thank you for your careful consideration. You are correcting that K2Cr2O7 solution could induce osmotic stress in oat seedlings. However, previous study indicates that a 200 µmol K2Cr2O7 solution may not cause significant osmotic stress in plants: “It is thought that osmotic stress occurs when the concentration of salt exceeds the threshold value of 10−3 M. Heavy metal ions seldom reach solution concentrations sufficient to cause osmotic disturbances in plants without previous symptoms of lethal toxicity” [16]. This might explain why similar studies have not mentioned considering osmotic stress caused by heavy metals[2, 17-20]. Therefore, we might not have accounted for the osmotic stress induced by K2Cr2O7 solution in our heavy metal stress experiments. However, your observation could indeed slightly affect the results. We will take your suggestion into account in future research to ensure more accurate results.
Comment 14: Section 2.10: Current best practice is two or more endogenous reference genes. Only AsEIF4A was selected, based on Yang et al. (2020; ref 56); however, those authors demonstrated that AsEIF4A and Heterogeneous nuclear ribonucleoprotein 27C (HNR) were best in tandem overall, and that "UBC21, an example of a four-copy duplicated RG, along with HNR, was identified as the most stable RG set in shoots and roots of oat seedlings." Why didn’t the authors follow this recommendation? Moreover, did the authors verify the stability of AsEIF4A in the tissues and conditions used? Response: Thank you for your question. We selected AsEIF4A as a single reference gene because Yang et al. demonstrated that it was stably expressed in all samples[21]. Additionally, previous studies on the Avena sativa gene family have effectively used this single reference gene [22-25]. In our study, we confirmed that AsEIF4A was stably expressed under five different abiotic stresses (heat, Cr, waterlogging, ABA-mediated, and wounding stresses). However, Yang et al. also suggested that combining reference genes improves accuracy. Therefore, we will take your suggestion into account and use multiple reference genes in future RT-qPCR experiments.
Comment 15: In addition, Yang et al. (2020; ref 56) used shoots (presumably including leaves). Do the authors mean shoots plus leaves or leaves only? What are the potential consequences of any difference? Response: Thank you for your question. Yang et al. collected samples from oat seedlings during the seeding stage, where the shoots consisted solely of leaves[21]. Therefore, the "shoots" mentioned in Yang et al.’s study are essentially the same as the "leaves" in our study.
Comment 16: The authors need to include the citations from which these publicly available data (SRR5097544–SRR5097549) and drought stress (SRR10718331–SRR10718339)) are taken or include the methods in a Supplementary file. Response: We have included the citation for the transcriptome data under salt stress (SRR5097544–SRR5097549) [26]. However, the transcriptome data under drought stress (SRR10718331–SRR10718339) have not been publicly published, and we sincerely apologize for this. As for the detailed information on Avena sativa under drought stress, we retrieved it from the NCBI website (https://www.ncbi.nlm.nih.gov/sra/SRR10718331 ) Line877: Add: “[26]”
Comment 17: Are the methods comparable to those presented in the current Materials and Methods? Response: Thank you for your question. The purpose of referencing the transcriptome studies was to explore potential expression patterns of the 12 AsAOX genes under salt and drought stress. Therefore, our study did not conduct RT-qPCR experiments under salt and drought stress to validate the results of other studies on Avena sativa. However, if one of the aims of our study were to validate others' transcriptome findings, your comment would have been extremely valuable. We apologize for any misunderstanding caused by the manuscript.
Comment 18: Were the plants comparable to those used in the current study? For example, young plant responses may not equate to older plant responses. Plant organ structure and developmental processes may influence how a plant responds to stimuli. Response: Thank you for your question. Both transcriptome studies used oat seedling leaves under salt and drought stress, similar to our study, which investigates the expression of AsAOX genes in leaves under various abiotic stresses. Therefore, their findings are highly relevant to our exploration of AOX gene expression under different stress conditions.
Comment 19: Lines 454-456: Please confirm 210-fold versus 210 times. They are mathematically distinct, resulting in drastically different values. Unfortunately, they are conflated in analyses such as this. Response: Thank you for your careful reading and valuable comments. After reviewing the sentence, I confirm that I meant "210-fold" in expression relative to the control (CK). Line521: Add: “-fold” Line522: Add: “-fold”
Comment 20: Section 4.4: The authors used 0-time samples as the control unless the signal was undetectable, in which case the subsequent sample was used as the control. A question arises as to circadian rhythms and whether any of the observed trends are related to time of day rather than stress. How have the authors controlled for circadian variation in the data? Response: Thank you for the insightful comment. To eliminate the influence of other factors on the experimental results, we included non-stressed oat seedlings as controls. We apologize for the oversight in not clearly describing this in the Materials and Methods section. The relative gene expression levels obtained under different abiotic stresses have already accounted for and excluded the effects of circadian rhythms. Sorry for the oversight in not providing a detailed description in the Materials and Methods section. We have now added the relevant details for your review. Line844-845: Add: “Additionally, we used non-stressed oat seedlings as controls to minimize the influence of other factors on the experimental results.”
Comment 21: Line 598: Was CRE defined before? If not, it should be defined here. Response: Thank you for your suggestion. We have added the detailed definition of CRE at this position: "Cis-regulatory elements exist in the gene side sequence, a piece of DNA sequence that can affect gene expression, and their role is to participate in the regulation of gene expression." Additionally, we have also included the full term for CRE at its first occurrence in the manuscript to aid in a better understanding of the concept. Line543-544: Add: “CREs exist in the gene side sequence, a piece of DNA sequence that can affect gene expression, and their role is to participate in the regulation of gene expression.” Line270: Replace: “CREs” with “cis-regulatory elements (CREs)”
Comment 22: Line 614: "Significantly" in the statistical sense or a qualitative sense? Response: Thank you for your question. In this context, the term "significantly" is used in a qualitative sense to describe the observed difference in relative gene expression levels between specific time points. It is not intended to imply statistical significance. We apologize for any confusion caused by this phrasing in the manuscript. We will revise the manuscript to clarify this distinction. Line361: Replace “significantly” with “notably” Line368: Replace “significant” with “notable” Line381: Replace “significantly” with “notably” Line519: Replace “significantly” with “notably” Line520: Replace “significantly” with “notably” Line525: Replace “significantly” with “notably” Line534: Replace “significantly” with “notably” Line562: Replace “significantly” with “notably” Line564: Replace “significantly” with “notably” Line576: Replace “significantly” with “notably” Line605: Replace “significantly” with “notably” Line607: Replace “significantly” with “notably” Line888: Replace “significant” with “notable”
Comment 23: Line 630: It is understood that the authors use "ABA stress" as shorthand for ABA-mediated responses, but it is jarring nonetheless. Response: Thank you for your suggestion. We have made the necessary revisions in the manuscript to improve clarity and eliminate any confusion regarding the term "ABA stress." Line29: Replace “ABA stress” with “ABA-mediated stress” Line361: Replace “ABA stress” with “ABA-mediated stress” Line550: Replace “ABA stress” with “ABA-mediated stress” Line556: Replace “ABA stress” with “ABA-mediated stress” Line559: Replace “ABA stress” with “ABA-mediated stress” Line561: Replace “ABA stress” with “ABA-mediated stress” Line564: Replace “ABA stress” with “ABA-mediated stress” Line582: Replace “ABA stress” with “ABA-mediated stress” Line583-584: Replace “ABA stress” with “ABA-mediated stress” Line859: Replace “ABA stress” with “ABA-mediated stress”
Comment 24: Line 634: Perhaps "stimulates" or "upregulates" rather than "enhances." Response: Thank you for your recommendation. We agree with your suggestion and have replaced the word "enhances" with "upregulates" in the revised manuscript, as it more accurately conveys the intended meaning. |
|
Line525: Replace “enhances” with “upregulates”
3. Reference |
Reviewer 2 Report
Comments and Suggestions for Authors
Liu, Boyang et al. Exploring Evolutionary Path and Abiotic Stress Responses through Genome-Wide Identification and Analysis of the Alternative Oxidase (AOX) Gene Family in Common Oat (Avena sativa)
The paper describes comprehensive studies regarding oat alternative oxidase (AOX) gene family which a common terminal oxidase in the electron transfer chain (ETC) of plants, plays a crucial role in stress resilience and plant growth and development. The paper has been well written as following: Regarding AOX gene family, 12 AsAOX (Avena sativa), 9 AiAOX (Avena insularis), and 4 AlAOX (Avena longiglumis) gene family members were identified and evolutionarily conservation among AOX gene families was demonstrated with the studies by phylogenetic, motif, domain, gene structure, and selective pressure analyses. 16 AsAOX segmental duplication pairs, indicating that segmental duplication, were also discovered, suggesting facilitation of the expansion of the AsAOX gene family. The evolution and expansion of the AsAOX gene family may have been driven by segmental duplications, chromosome polyploidization, and gene structural variations. Some AsAOX genes, such as AsAOX11, exhibit significant tissue specificity between leaves and roots. Additionally, AsAOX2, 4, 5, 8, 11, and 12 may play roles in resisting various abiotic stresses.
Minor suggestion of revision:
Introduction:
The first paragraph describing general issues is not necessarily necessary, but it would be better to mention the content about AOX and the need for its study.
Discussion:  
Is there any evolutionary significance on AOX in comparing the three species of Avena and also among Poaceae species?
There should be a mention of prospects for this study on AOX gene family.
Author Response
- Point-by-point response to Comments and Suggestions for Authors
Comment 1: Introduction: The first paragraph describing general issues is not necessarily necessary, but it would be better to mention the content about AOX and the need for its study.
Response: Thank you for your valuable suggestions after thoroughly reviewing the manuscript. We apologize for dedicating a significant portion of the first paragraph to general issues and providing limited discussion about AOX. In response to your comments, we have reduced the content on general issues in the first paragraph and included more information about AOX. Additionally, we have enriched the second paragraph of the introduction with further background on AOX.
However, the logical structure of this study's introduction starts by addressing the general abiotic stresses that plants face and gradually introduces the role of the AOX gene in mitigating these stresses. The purpose of the first paragraph is to provide readers with a broad understanding of the research background and the physiological responses of plants to various stresses. The second paragraph then goes on to elaborate on the importance of the AOX gene family, its function, and its involvement in stress response, laying the foundation for the necessity of this research. Finally, the third paragraph delves into the current research on AOX genes in Avena sativa, highlighting the research gaps and defining the objectives and significance of this study. Thus, according to the logical flow of the introduction, the detailed content about AOX and the rationale for this study could be fully addressed in the second and third paragraphs to maintain the coherence and logical structure of the introduction. We sincerely apologize if our manuscript has inadvertently misled you.
Line35-50: Introduction: Revise the first paragraph and add some content regarding the AOX respiratory pathway.
“Throughout their lifecycle, higher plants face a variety of abiotic stresses, such as chromium contamination, extreme temperatures, waterlogging, and pest damage, which can lead to irreversible cellular damage and significant yield losses[1-7]. Abscisic acid (ABA), a key plant hormone, plays a crucial role in regulating many physiological responses to these stresses. To mitigate these challenges, plants have evolved complex mechanisms involving gene expression and regulatory networks[3, 8]. Key gene families, including trihelix, HSP, glyoxalase, and AOX, are essential for stress resistance[4, 9-11]. Additionally, plant respiration is tightly controlled by metabolic networks that allow plants to adjust their metabolism in response to internal cellular demands and external environmental stresses. Notably, higher plants, algae, and most fungi possess an alternative respiratory pathway in their mitochondria, which is resistant to cyanide and bypasses the cytochrome pathway, ultimately reducing oxygen to water[12]. The alternative oxidase (AOX) is a key component of this pathway and is critical for its proper function under stress conditions.”
Line52-55: Add: “AOX is an integral membrane protein located in the mitochondrial inner membrane. It can regulate the levels of potential mitochondrial signaling molecules, such as superoxide, nitric oxide, and key redox pairs[13].”
Line59-61: Add: “Therefore, AOX genes respond to various stress conditions, particularly those that affect ROS levels, such as changes in temperature and light intensity, as well as salt, drought, and osmotic stress [14-17].”
Line65-67: Add: “Due to its involvement in nearly all types of stress responses, the AOX gene is considered a valuable marker for stress tolerance in crop breeding, aiming to produce plants with high resilience to various environmental conditions[18].”
Line70-72: Add” The AOX1 subfamily is represented by multiple gene copies in many plants, while the AOX2 subfamily shows relatively fewer expansions[19].”
Comment 2: Discussion: Is there any evolutionary significance on AOX in comparing the three species of Avena and also among Poaceae species?
Response: Thank you for your valuable suggestions after thoroughly reviewing the discussion section of the manuscript. We apologize for the previous lack of discussion on the evolutionary significance of AOX among the three Avena species and within the Poaceae family. In response to your comments, we have added a new section (3.3 Exploration of Evolutionary Conservation and Divergence) to the Discussion, which addresses the evolutionary conservation and divergence of AOX among the three Avena species and within the Poaceae family, using three key Poaceae model species—Oryza sativa, Triticum aestivum, and Brachypodium distachyon—as examples.
Line461-502: Add:
“3.3 Exploration of Evolutionary Conservation and Divergence
AiAOXs and AlAOXs are considered ancestral genes to AsAOX genes[20]. Most of the genes exhibit strong evolutionary conservation, as reflected in the similar motif composition, gene structure, evolutionary branches, and orthologous gene pairs across multiple AOX genes(Figure 4, and Table S3). Based on our findings, AsAOX11, AsAOX5, and AsAOX12 are potential candidate genes with functions related to environmental stress resistance. Since AiAOX4 and AsAOX11 show a high degree of evolutionary conservation, it is likely that the potential functions of AsAOX11 were inherited from its ancestral gene, AiAOX4. Similarly, we infer that the potential functions of AsAOX5 and AsAOX12 in resisting salt and drought stress likely originated from their respective ancestral genes, AiAOX6 and AiAOX3. However, some AsAOX genes have diverged from their ancestral AiAOX genes in terms of motif composition, or gene structure.We suggest that AsAOX2, AsAOX4, and AsAOX8 may also have potential functions in stress resistance. However, AsAOX2 shows differences in motif composition, and gene structure compared to AiAOX5, AiAOX6, and AiAOX7, which are located on the same evolutionary branch, indicating functional divergence. Similarly, AsAOX4 and AsAOX8 have also diverged evolutionarily from several AiAOX genes on the same branch. Therefore, the potential stress-resistance functions of AsAOX2, AsAOX4, and AsAOX8 may not have been inherited from their ancestral genes, but rather, arose from gene fragment recombination during evolution[21]. Overall, there is both strong evolutionary conservation and divergence between the AsAOX genes and their ancestral AiAOX and AlAOX genes, contributing to the acquisition of potential functions by certain AsAOX genes.
As an important member of the Poaceae family, the evolutionary conservation of AsAOX genes in Poaceae has yet to be fully elucidated. Therefore, we explored the evolutionary relationships of the AOX gene family among A. sativa and three key Poaceae model species—Oryza sativa, Triticum aestivum, and Brachypodium distachyon (Figure S6). Our findings revealed both evolutionary conservation and divergence in AOX genes among A. sativa and these Poaceae model species. For example, AsAOX1, AsAOX3, and AsAOX6 share high similarity in motif composition and gene structure with OsAOX3, TaAOX1-3, and BdAOX1, which are located on the same evolutionary branch, indicating strong evolutionary conservation. However, other AsAOX, OsAOX, TaAOX, and BdAOX genes exhibit significant evolutionary divergence, characterized by differences in motif composition or gene structure. Thus, there is considerable evolutionary divergence and limited conservation of AOX genes among A. sativa and these three important Poaceae model species.
Additionally, we found that AsAOX2 shows substantial evolutionary divergence not only within the 12 AsAOX genes but also when compared to AiAOX, AlAOX, OsAOX, TaAOX, and BdAOX genes. This pronounced evolutionary divergence may explain its superior performance in our RT-qPCR experiments under the five abiotic stress conditions tested in this study(Figure 9).”
Comment 3: Discussion: here should be a mention of prospects for this study on AOX gene family.
Response: Thank you for your valuable suggestions after thoroughly reviewing the Discussion of the manuscript. We apologize for the previous lack of discussion on the prospects for this study on the AOX gene family. In response to your comments, we have added a new section (3.8 Prospects for the AsAOX Gene Family) to the Discussion, which elaborates on the future prospects for the AsAOX gene family.
Line610-618: Add:
“3.8 Prospects for AsAOX gene family
In conclusion, our study identifies AsAOX2, 4, 5, 8, 11, and 12 as potential genes for resisting various abiotic stresses. Future research should validate these genes' functions in A. sativa and other important crops to improve stress tolerance. Additionally, AsAOX11 shows strong tissue specificity in roots and leaves under multiple stresses, highlighting the need to investigate its tissue-specific regulatory mechanisms. Furthermore, AsAOX3's significant ABA response despite having few ABREs suggests it is regulated by a complex CREs network. Future studies should explore these regulatory networks to better understand AsAOX3's role in ABA-mediated stress responses.”

Round 2
Reviewer 1 Report
Comments and Suggestions for Authors
Much improved over the original manuscript.